# Brian 2, an intuitive and efficient neural simulator

**Marcel Stimberg[1]\*, Romain Brette[1][†], Dan FM Goodman[2][†]**

[1]Sorbonne Université, INSERM, CNRS, Institut de la Vision, Paris, France;
[2]Department of Electrical and Electronic Engineering, Imperial College London, London, United Kingdom

**Abstract** Brian 2 allows scientists to simply and efficiently simulate spiking neural network models. These models can feature novel dynamical equations, their interactions with the environment, and experimental protocols. To preserve high performance when defining new models, most simulators offer two options: low-level programming or description languages. The first option requires expertise, is prone to errors, and is problematic for reproducibility. The second option cannot describe all aspects of a computational experiment, such as the potentially complex logic of a stimulation protocol. Brian addresses these issues using runtime code generation. Scientists write code with simple and concise high-level descriptions, and Brian transforms them into efficient low-level code that can run interleaved with their code. We illustrate this with several challenging examples: a plastic model of the pyloric network, a closed-loop sensorimotor model, a programmatic exploration of a neuron model, and an auditory model with real-time input.
DOI: https://doi.org/10.7554/eLife.47314.001

**\*For correspondence:**
marcel.stimberg@inserm.fr

[†]These authors contributed equally to this work

**Competing interests:** The authors declare that no competing interests exist.

## Introduction

Neural simulators are increasingly used to develop models of the nervous system, at different scales and in a variety of contexts (*Brette et al., 2007*). These simulators generally have to find a trade-off between performance and the flexibility to easily define new models and computational experiments. Brian 2 is a complete rewrite of the Brian simulator designed to solve this apparent dichotomy using the technique of code generation. The design is based around two fundamental ideas. Firstly, it is equation based: defining new neural models should be no more difficult than writing down their equations. Secondly, the computational experiment is fundamental: the interactions between neurons, environment and experimental protocols are as important as the neural model itself. We cover these points in more detail in the following paragraphs.

Popular tools for simulating spiking neurons and networks of such neurons are NEURON (*Carnevale and Hines, 2006*), GENESIS (*Bower and Beeman, 1998*), NEST (*Gewaltig and Diesmann, 2007*), and Brian (*Goodman and Brette, 2008*; *Goodman and Brette, 2009*; *Goodman and Brette, 2013*). Most of these simulators come with a library of standard models that the user can choose from. However, we argue that to be maximally useful for research, a simulator should also be designed to facilitate work that goes beyond what is known at the time that the tool is created, and therefore enable the user to investigate new mechanisms. Simulators take widely different approaches to this issue. For some simulators, adding new mechanisms requires specifying them in a low-level programming language such as C++, and integrating them with the simulator code (e.g. NEST). Amongst these, some provide domain-specific languages, for example NMODL (*Hines and Carnevale, 2000*, for NEURON) or NESTML (*Plotnikov et al., 2016*, for NEST), and tools to transform these descriptions into compiled modules that can then be used in simulation scripts. Finally, the Brian simulator has been built around mathematical model descriptions that are part of the simulation script itself.

**eLife digest** Simulating the brain starts with understanding the activity of a single neuron. From there, it quickly gets very complicated. To reconstruct the brain with computers, neuroscientists have to first understand how one brain cell communicates with another using electrical and chemical signals, and then describe these events using code. At this point, neuroscientists can begin to build digital copies of complex neural networks to learn more about how those networks interpret and process information.

To do this, computational neuroscientists have developed simulators that take models for how the brain works to simulate neural networks. These simulators need to be able to express many different models, simulate these models accurately, and be relatively easy to use. Unfortunately, simulators that can express a wide range of models tend to require technical expertise from users, or perform poorly; while those capable of simulating models efficiently can only do so for a limited number of models. An approach to increase the range of models simulators can express is to use so-called 'model description languages'. These languages describe each element within a model and the relationships between them, but only among a limited set of possibilities, which does not include the environment. This is a problem when attempting to simulate the brain, because a brain is precisely supposed to interact with the outside world.

Stimberg et al. set out to develop a simulator that allows neuroscientists to express several neural models in a simple way, while preserving high performance, without using model description languages. Instead of describing each element within a specific model, the simulator generates code derived from equations provided in the model. This code is then inserted into the computational experiments. This means that the simulator generates code specific to each model, allowing it to perform well across a range of models. The result, Brian 2, is a neural simulator designed to overcome the rigidity of other simulators while maintaining performance.

Stimberg et al. illustrate the performance of Brian 2 with a series of computational experiments, showing how Brian 2 can test unconventional models, and demonstrating how users can extend the code to use Brian 2 beyond its built-in capabilities.

DOI: https://doi.org/10.7554/eLife.47314.002

Another approach to model definitions has been established by the development of simulator-independent markup languages, for example NeuroML/LEMS (*Gleeson et al., 2010*; *Cannon et al., 2014*) and NineML (*Raikov et al., 2011*). However, markup languages address only part of the problem. A computational experiment is not fully specified by a neural model: it also includes a particular experimental protocol (set of rules defining the experiment), for example a sequence of visual stimuli. Capturing the full range of potential protocols cannot be done with a purely declarative markup language, but is straightforward in a general purpose programming language. For this reason, the Brian simulator combines the model descriptions with a procedural, computational experiment approach: a simulation is a user script written in Python, with models described in their mathematical form, without any reference to predefined models. This script may implement arbitrary protocols by loading data, defining models, running simulations and analysing results. Due to Python's expressiveness, there is no limit on the structure of the computational experiment: stimuli can be changed in a loop, or presented conditionally based on the results of the simulation, etc. This flexibility can only be obtained with a general-purpose programming language and is necessary to specify the full range of computational experiments that scientists are interested in.

While the procedural, equation-oriented approach addresses the issue of flexibility for both the modelling and the computational experiment, it comes at the cost of reduced performance, especially for small-scale models that do not benefit much from vectorisation techniques (*Brette and Goodman, 2011*). The reduced performance results from the use of an interpreted language to implement arbitrary models, instead of the use of pre-compiled code for a set of previously defined models. Thus, simulators generally have to find a trade-off between flexibility and performance, and previous approaches have often chosen one over the other. In practice, this makes computational experiments that are based on non-standard models either difficult to implement or slow to perform. We will describe four case studies in this article: exploring unconventional plasticity rules for a

small neural circuit (case study 1, *Figure 1*, *Figure 2*); running a model of a sensorimotor loop (case study 2, *Figure 3*); determining the spiking threshold of a complex model by bisection (case study 3, *Figure 4*, *Figure 5*); and running an auditory model with real-time input from a microphone (case study 4, *Figure 6*, *Figure 7*).

Brian 2 solves the performance-flexibility trade-off using the technique of code generation (*Goodman, 2010*; *Stimberg et al., 2014*; *Blundell et al., 2018*). The term code generation here refers to the process of automatically transforming a high-level user-defined model into executable code in a computationally efficient low-level language, compiling it in the background and running it without requiring any actions from the user. This generated code is inserted within the flow of the simulation script, which makes it compatible with the procedural approach. Code generation is not only used to run the models but also to build them, and therefore also accelerates stages such as synapse creation. The code generation framework has been designed to be extensible on several levels. On a general level, code generation targets can be added to generate code for other architectures, for example graphical processing units, from the same simulation description. On a more specific level, new functionality can be added by providing a small amount of code written in the target language, for example to connect the simulation to an input device. Implementing this solution in a way that is transparent to the user requires solving important design and computational problems, which we will describe in the following.

## Materials and methods

### Design and implementation

We will explain the key design decisions by starting from the requirements that motivated them. Note that from now on we will use the term 'Brian' as referring to its latest version, that is Brian 2, and only use 'Brian 1' and 'Brian 2' when discussing differences between them.

Before discussing the requirements, we start by motivating the choice of programming language. Python is a high-level language, that is, it is abstracted from machine level details and highly readable (indeed, it is often described as 'executable pseudocode'). In this sense, it is higher level than C++, for example, which in this article we will refer to as a low-level language (since we will not need to refer to even lower level languages such as assembly language). The use of a high-level language is important for scientific software because the majority of scientists are not trained programmers, and high-level languages are generally easier to learn and use, and lead to shorter code that is easier to debug. This last point, and the fact that Python is a very popular general purpose programming language with excellent built-in and third party tools, is also important for reducing development time, enabling the developers to be more efficient. It is now widely recognised that Python is well suited to scientific software, and it is commonly used in computational neuroscience (*Davison et al., 2009*; *Muller et al., 2015*). Note that expert level Python knowledge is not necessary for using Brian or the Python interfaces for other simulators.

We now move on to the major design requirements.

1. Users should be able to easily define non-standard models, which may include models of neurons and synapses but also of other aspects such as muscles and environment. This is made possible by an equation-oriented approach, that is models are described by mathematical equations. We first focus on the design at the *mathematical level*, and we illustrate with two unconventional models: a model of intrinsic plasticity in the pyloric network of the crustacean stomatogastric ganglion (case study 1, *Figure 1*, *Figure 2*), and a closed-loop sensorimotor model of ocular movements (case study 2, *Figure 3*).
2. Users should be able to easily implement a complete computational experiment in Brian. Models must interact with a general control flow, which may include stimulus generation and various operations. This is made possible by taking a procedural approach to defining a complete computational experiment, rather than a declarative model definition, allowing users to make full use of the generality of the Python language. In the section on the *computational experiment level*, we demonstrate the interaction between a general control flow expressed in Python and the simulation run in a case study that uses a bisection algorithm to determine a neuron's firing threshold as a function of sodium channel density (case study 3, *Figure 4*, *Figure 5*).

3. Computational efficiency. Often, computational neuroscience research is limited more by the scientist's time spent designing and implementing models, and analysing results, rather than the simulation time. However, there are occasions where high computational efficiency is necessary. To achieve high performance while preserving maximum flexibility, Brian generates code from user-defined equations and integrates it into the simulation flow.

4. Extensibility: no simulator can implement everything that any user might conceivably want, but users shouldn't have to discard the simulator entirely if they want to go beyond its built-in capabilities. We therefore provide the possibility for users to extend the code either at a high or low level. We illustrate these last two requirements at the *implementation level* with a case study of a model of pitch perception using real-time audio input (case study 4, *Figure 6*, *Figure 7*).

In this section, we give a high level overview of the major decisions. A detailed analysis of the case studies and the features of Brian they use can be found in Appendix 1. Source code for the case studies has been deposited in a repository at https://github.com/brian-team/brian2_paper_examples (*Stimberg et al., 2019a*; copy archived at https://github.com/elifesciences-publications/brian2_paper_examples).

## Mathematical level
### Case study 1: Pyloric network

We start with a case study of a model of the pyloric network of the crustacean stomatogastric ganglion (*Figure 1a*), adapted and simplified from earlier studies (*Golowasch et al., 1999*; *Prinz et al., 2004*; *Prinz, 2006*; *O'Leary et al., 2014*). This network has a small number of well-characterised neuron types – anterior burster (AB), pyloric dilator (PD), lateral pyloric (LP), and pyloric (PY) neurons – and is known to generate a stereotypical triphasic motor pattern (*Figure 1b–c*). Following previous studies, we lump AB and PD neurons into a single neuron type (AB/PD) and consider a circuit with one neuron of each type. The neurons in this circuit have rebound and bursting properties. We

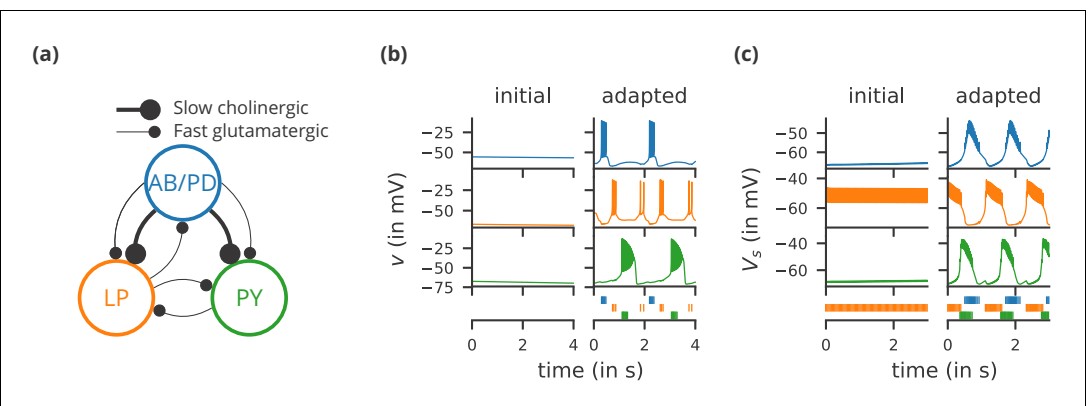

**Figure 1.** Case study 1: A model of the pyloric network of the crustacean stomatogastric ganglion, inspired by several modelling papers on this subject (*Golowasch et al., 1999*; *Prinz et al., 2004*; *Prinz, 2006*; *O'Leary et al., 2014*). (a) Schematic of the modelled circuit (after *Prinz et al., 2004*). The pacemaker kernel is modelled by a single neuron representing both anterior burster and pyloric dilator neurons (AB/PD, blue). There are two types of follower neurons, lateral pyloric (LP, orange), and pyloric (PY, green). Neurons are connected via slow cholinergic (thick lines) and fast glutamatergic (thin lines) synapses. (b) Activity of the simulated neurons. Membrane potential is plotted over time for the neurons in (a), using the same colour code. The bottom row shows their spiking activity in a raster plot, with spikes defined as excursions of the membrane potential over −20 mV. In the left column ('initial'), activity is shown for 4 s after an initial settling time of 2.5 s. The right column ('adapted') shows the activity with fully adapted conductances (see text for details) after an additional time of 49 s. (c) Activity of the simulated neurons of a biologically detailed version of the circuit shown in (a), following (*Golowasch et al., 1999*). All conventions as in (b), except for showing 3 s of activity after a settling time of 0.5 s ('initial'), and after an additional time of 24 s ('adapted'). Also note that the biologically detailed model consists of two coupled compartments, but only the membrane potential of the somatic compartment ($V_s$) is shown here.
DOI: https://doi.org/10.7554/eLife.47314.003

```
1   from brian2 import *
2   defaultclock.dt = 0.01*ms;
3   Delta_T = 17.5*mV          ; v_T = -40*mV        ; tau = 2*ms          ; tau_adapt = .02*second
4   tau_Ca = 150*ms            ; tau_x = 2*second    ; v_r = -68*mV        ; tau_z = 5*second
5   a = 1/Delta_T**3           ; b = 3/Delta_T**2    ; c = 1.2*nA          ; d = 2.5*nA/Delta_T**2
6   C = 60*pF                  ; S = 2*nA/Delta_T    ; G = 28.5*nS
7   eqs = '''
8   dv/dt = (Delta_T*g*(-a*(v - v_T)**3 + b*(v - v_T)**2) + w - x - I_fast - I_slow)/C : volt
9   dw/dt = (c - d*(v - v_T)**2 - w)/tau : amp
10  dx/dt = (s*(v - v_r) - x)/tau_x : amp
11  s = S*(1 - tanh(z)) : siemens
12  g = G*(1 + tanh(z)) : siemens
13  dCa/dt = -Ca/tau_Ca : 1
14  dz/dt = tanh(Ca - Ca_target)/tau_z : 1
15  I_fast : amp
16  I_slow : amp
17  Ca_target : 1 (constant)
18  label : integer (constant)
19  '''
20  ABPD, LP, PY = 0, 1, 2
21  circuit = NeuronGroup(3, eqs, threshold='v>-20*mV', refractory='v>-20*mV', reset='Ca += 0.1',
22                        method='rk2')
23  circuit.label = [ABPD, LP, PY]
24  circuit.v = v_r
25  circuit.w = '-5*nA*rand()'
26  circuit.z = 'rand()*0.2 - 0.1'
27  circuit.Ca_target = [0.048, 0.0384, 0.06]
28
29  s_fast = 0.2/mV; V_fast = -50*mV; E_syn = -75*mV
30  eqs_fast = '''
31  g_fast : siemens (constant)
32  I_fast_post = g_fast*(v_post - E_syn)/(1+exp(s_fast*(V_fast-v_pre))) : amp (summed)
33  '''
34  fast_synapses = Synapses(circuit, circuit, model=eqs_fast)
35  fast_synapses.connect('label_pre != label_post and not (label_pre == PY and label_post == ABPD)')
36  fast_synapses.g_fast['label_pre == ABPD and label_post == LP'] = 0.015*uS
37  fast_synapses.g_fast['label_pre == ABPD and label_post == PY'] = 0.005*uS
38  fast_synapses.g_fast['label_pre == LP and label_post == ABPD'] = 0.01*uS
39  fast_synapses.g_fast['label_pre == LP and label_post == PY']   = 0.02*uS
40  fast_synapses.g_fast['label_pre == PY and label_post == LP']   = 0.005*uS
41
42  s_slow = 1/mV; V_slow = -55*mV; k_1 = 1/ms
43  eqs_slow = '''
44  k_2 : 1/second (constant)
45  g_slow : siemens (constant)
46  I_slow_post = g_slow*m_slow*(v_post-E_syn) : amp (summed)
47  dm_slow/dt = k_1*(1-m_slow)/(1+exp(s_slow*(V_slow-v_pre))) - k_2*m_slow : 1 (clock-driven)
48  '''
49  slow_synapses = Synapses(circuit, circuit, model=eqs_slow, method='exact')
50  slow_synapses.connect('label_pre == ABPD and label_post != ABPD')
51  slow_synapses.g_slow['label_post == LP'] = 0.025*uS
52  slow_synapses.k_2['label_post == LP']    = 0.03/ms
53  slow_synapses.g_slow['label_post == PY'] = 0.015*uS
54  slow_synapses.k_2['label_post == PY']    = 0.008/ms
55
56  run(59.5*second)
```

**Figure 2.** Case study 1: A model of the pyloric network of the crustacean stomatogastric ganglion. Simulation code for the model shown in *Figure 1a*, producing the circuit activity shown in *Figure 1b*.

DOI: https://doi.org/10.7554/eLife.47314.004

The following figure supplement is available for figure 2:

**Figure supplement 1.** Simulation code for the more biologically detailed model of the circuit shown in *Figure 1a*, producing the circuit activity shown in *Figure 1c*.

DOI: https://doi.org/10.7554/eLife.47314.005

model this using a variant of the model proposed by *Hindmarsh and Rose (1984)*, a three-variable model exhibiting such properties. We make this choice only for simplicity: the biophysical equations originally used in *Golowasch et al. (1999)* can be used instead (see *Figure 2—figure supplement 1*).

Although this model is based on widely used neuron models, it has the unusual feature that some of the conductances are regulated by activity as monitored by a calcium trace. One of the first design requirements of Brian, then, is that non-standard aspects of models such as this should be as easy to implement in code as they are to describe in terms of their mathematical equations. We briefly summarise how it applies to this model (see Appendix 1 and *Stimberg et al., 2014* for more detail). The three-variable underlying neuron model is implemented by writing its differential equations directly in standard mathematical form (*Figure 2*, l. 8–10). The calcium trace increases at each spike (l. 21; defined by a discrete event triggered after a spike, `reset='Ca + = 0.1'`) and then decays (l. 13; again defined by a differential equation). A slow variable $z$ tracks the difference of this calcium trace to a neuron-type-specific target value (l. 14) which then regulates the conductances $s$ and $g$ (l. 11–12).

Not only the neuron model but also their connections are non-standard. Neurons are connected together by nonlinear graded synapses of two different types, slow and fast (l. 29–54). These are unconventional synapses in that the synaptic current has a graded dependence on the pre-synaptic action potential and a continuous effect rather than only being triggered by pre-synaptic action potentials (*Abbott and Marder, 1998*). A key design requirement of Brian was to allow for the same expressivity for synaptic models as for neuron models, which led us to a number of features that allow for a particularly flexible specification of synapses in Brian. Firstly, we allow synapses to have dynamics defined by differential equations in precisely the same way as neurons. In addition to the usual role of triggering instantaneous changes in response to discrete neuronal events such as spikes, synapses can directly and continuously modify neuronal variables allowing for a very wide range of synapse types. To illustrate this, for the slow synapse, we have a synaptic variable (`m_slow`) that evolves according to a differential equation (l. 47) that depends on the pre-synaptic membrane potential (`v_pre`). The effect of this synapse is defined by setting the value of a post-synaptic neuron current (`I_slow`) in the definition of the synapse model (l. 46; referred to there as `I_slow_post`). The keyword (`summed`) in the equation specifies that the post-synaptic neuron variable is set using the summed value of the expression across all the synapses connected to it. Note that this mechanism also allows Brian to be used to specify abstract rate-based neuron models in addition to biophysical graded synapse models.

The model is defined not only by its dynamics, but also the values of parameters and the connectivity pattern of synapses. The next design requirement of Brian was that these essential elements of specifying a model should be equally flexible and readable as the dynamics. In this case, we have added a label variable to the model that can take values ABPD, LP or PY (l. 18, 20, 23) and used this label to set up the initial values (l. 36–40, 51–54) and connectivity patterns (l. 35, 50). Human readability of scripts is a key aspect of Brian code, and important for reproducibility (which we will come back to in the Discussion). We highlight line 35 to illustrate this. We wish to have synapses between all neurons of different types but not of the same type, except that we do not wish to have synapses from PY neurons to AB/PD neurons. Having set up the labels, we can now express this connectivity pattern with the expression `'label_pre!=label_post and not (label_pre == PY and label_post == ABPD)'`. This example illustrates one of the many possibilities offered by the equation-oriented approach to concisely express connectivity patterns (for more details see Appendix 1 and *Stimberg et al., 2014*).

## Comparison to other approaches

In this case study, we have shown how a non-standard neural network, with graded synapses and adapting conductances, can be described in the Brian simulator. How could such a model be implemented with one of the other approaches described previously? We will briefly discuss this by focussing on implementations of the graded synapse model. One approach is to directly write an implementation of the model in a programming language like C++, without the use of any simulation software. While this requires significant technical skill, it allows for complete freedom in the definition of the model itself. This was the approach taken for a study that ran 20 million parametrised

instances of the same pyloric network model (*Günay and Prinz, 2010*). The increased effort of writing the simulation was offset by reusing the same model an extremely large number of times. An excerpt of this code is shown in *Appendix 3—figure 1c*. Note that unless great care is taken, this approach may lead to a very specific implementation of the model that is not straightforward to adapt for other purposes. With such long source code (3510 lines in this case) it is also difficult to check that there are no errors in the code, or implicit assumptions that deviate from the description (as in, for example *Hathway and Goodman, 2018*; *Pauli et al., 2018*).

Another approach for describing model components such as graded synapses is to use a description language such as LEMS/NeuroML2. If the specific model has already been added as a 'core type', then it can be readily referenced in the description of the model (*Appendix 3—figure 2a*). If not, then the LEMS description can be used to describe it (*Appendix 3—figure 2a*). Such a description is on a similar level of abstraction as the Brian description, but somewhat more verbose (although this may be reduced by using a library such as PyLEMS to create the description; *Vella et al., 2014*).

If the user chooses to use the NEURON simulator to simulate the model, then a new synaptic mechanism can be added using the NMODL language (*Appendix 3—figure 2b*). However, for the user this requires learning a new, idiosyncratic language, and detailed knowledge about simulator internals, for example the numerical solution of equations. Other simulators, such as NEST, are focussed on discrete spike-based interactions and currently do not come with models of graded synapses, and such models are not yet supported by its description language NESTML. Leveraging the infrastructure added for gap-junctions (*Hahne et al., 2015*) and rate models (*Hahne et al., 2017*), the NEST simulator could certainly integrate such models in principle but in practice this may not be feasible without direct support from the NEST team.

## Case study 2: Ocular model

The second example is a closed-loop sensorimotor model of ocular movements (used for illustration and not intended to be a realistic description of the system), where the eye tracks an object (*Figure 3a,b*). Thus, in addition to neurons, the model also describes the activity of ocular muscles and the dynamics of the stimulus. Each of the two antagonistic muscles is modelled mechanically as an elastic spring with some friction, which moves the eye laterally.

The next design requirement of Brian was that it should be both possible and straightforward to define non-neuronal elements of a model, as these are just as essential to the model as a whole, and the importance of connecting with these elements is often neglected in neural simulators. We will come back to this requirement in various forms over the next few case studies, but here we emphasise how the mechanisms for specifying arbitrary differential equations can be re-used for non-neuronal elements of a simulation.

The position of the eye follows a second-order differential equation, with resting position $x_0$, the difference in resting positions of the two muscles (*Figure 3c*, l. 4–5). The stimulus is an object that moves in front of the eye according to a stochastic process (l. 7–8). Muscles are controlled by two motoneurons (l. 11–13), for which each spike triggers a muscular 'twitch'. This corresponds to a transient change in the resting position $x_0$ of the eye in either direction, which then decays back to zero (l. 6, 15).

Retinal neurons receive a visual input, modelled as a Gaussian function of the difference between the neuron's preferred position and the actual position of the object, measured in retinal coordinates (l. 21). Thus, the input to the neurons depends on dynamical variables external to the neuron model. This is a further illustration of the design requirement above that we need to include non-neuronal elements in our model specifications. In this case, to achieve this we link the variables in the eye model with the variables in the retina model using the `linked_var` function (l. 4, 7, 23–24, 28–29).

Finally, we implement a simple feedback mechanism by having retinal neurons project onto the motoneuron controlling the contralateral muscle (l. 33), with a strength proportional to their eccentricity (l. 36): thus, if the object appears on the edge of the retina, the eye is strongly pulled towards the object; if the object appears in the centre, muscles are not activated. This simple mechanism allows the eye to follow the object (*Figure 3b*), and the code illustrates the previous design requirement that the code should reflect the mathematical description of the model.

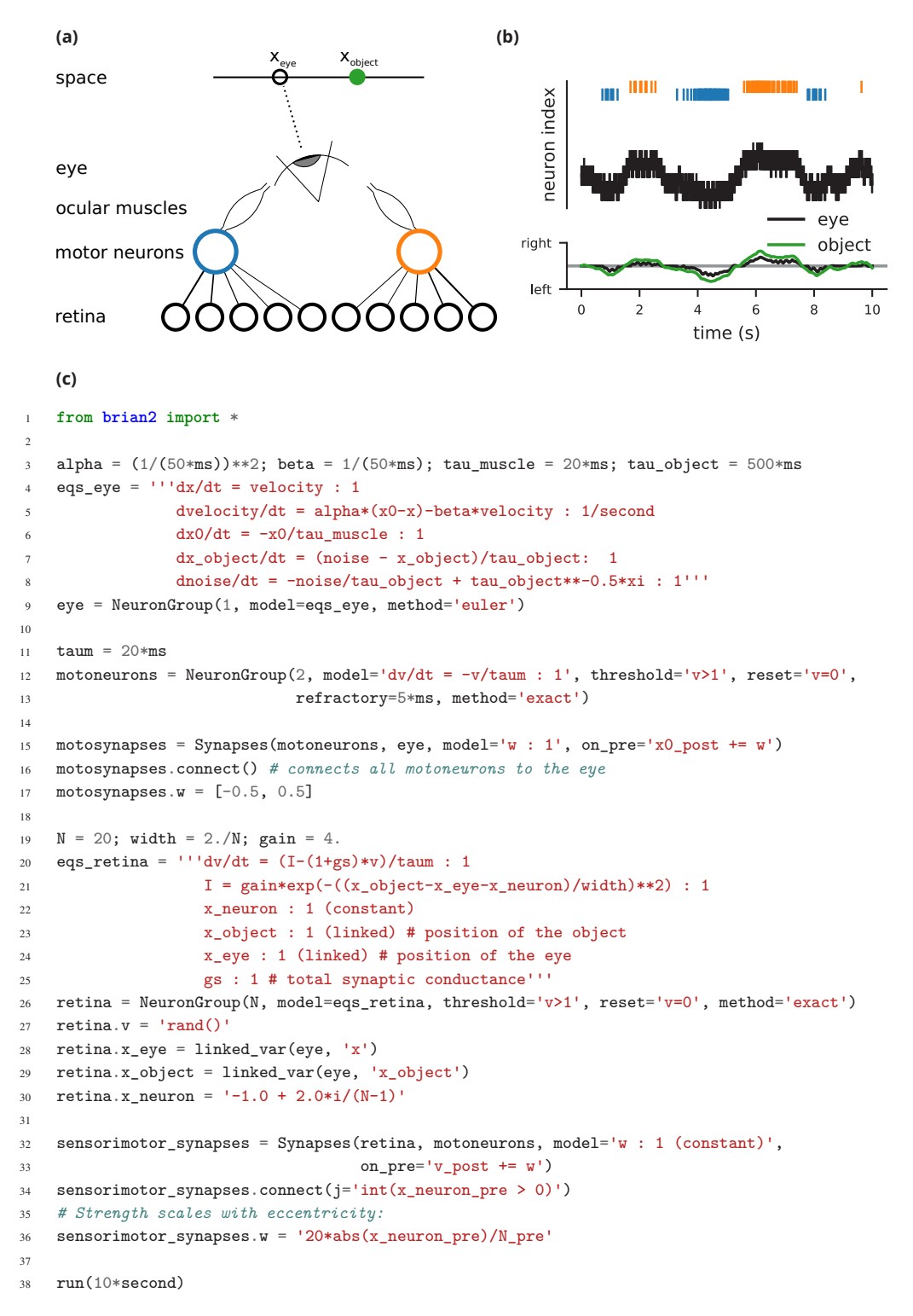

**Figure 3.** Case study 2: Smooth pursuit eye movements. (a) Schematics of the model. An object (green) moves along a line and activates retinal neurons (bottom row; black) that are sensitive to the relative position of the object to the eye. Retinal neurons activate two motor neurons with weights depending on the eccentricity of their preferred position in space. Motor neurons activate the ocular muscles responsible for turning the eye. (b) Top: Simulated activity of the sensory neurons (black), and the left (blue) and right (orange) motor neurons. Bottom: Position of the eye (black) and the stimulus (green). (c) Simulation code.

DOI: https://doi.org/10.7554/eLife.47314.006

## Comparison to other approaches

The remarks we made earlier regarding the graded synapse in case study one mostly apply here as well. For LEMS/NeuroML2, both motor neurons and the environment could be modelled with a LEMS description. Similarly, a simulation with NEURON would require NMODL specifications of both models, using its `POINTER` mechanism (see *Appendix 3—figure 2*) to link them together. Since NEST's modelling language NESTML does not allow for the necessary continuous interaction between a single environment and multiple neurons, implementing this model would be a major effort and require writing code in C++ and detailed knowledge of NEST's internal architecture.

## Computational experiment level

The mathematical model descriptions discussed in the previous section provide only a partial description of what we might call a 'computational experiment'. Let us consider the analogy to an electrophysiological experiment: for a full description, we would not only state the model animal, the cell type and the preparation that was investigated, but also the stimulation and analysis protocol. In the same way, a full description of a computational experiment requires not only a description of the neuron and synapse models, but also information such as how input stimuli are generated, or what sequence of simulations is run. Some examples of computational experimental protocols would include: threshold finding (discussed in detail below) where the stimulus on the next trial depends on the outcome of the current trial; generalisations of this to potentially very complex closed-loop experiments designed to determine the optimal stimuli for a neuron (e.g. *Edin et al., 2004*); models including a complex simulated environment defined in an external package (e.g. *Voegtlin, 2011*); or models with plasticity based on an error signal that depends on the global behaviour of the network (e.g. *Stroud et al., 2018*; *Zenke and Ganguli, 2018*). Capturing all these potential protocols in a purely descriptive format (one that is not Turing complete) is impossible by definition, but it can be easily expressed in a programming language with control structures such as loops and conditionals. The Brian simulator allows the user to write complete computational experimental protocols that include both the model description and the simulation protocol in a single, readable Python script.

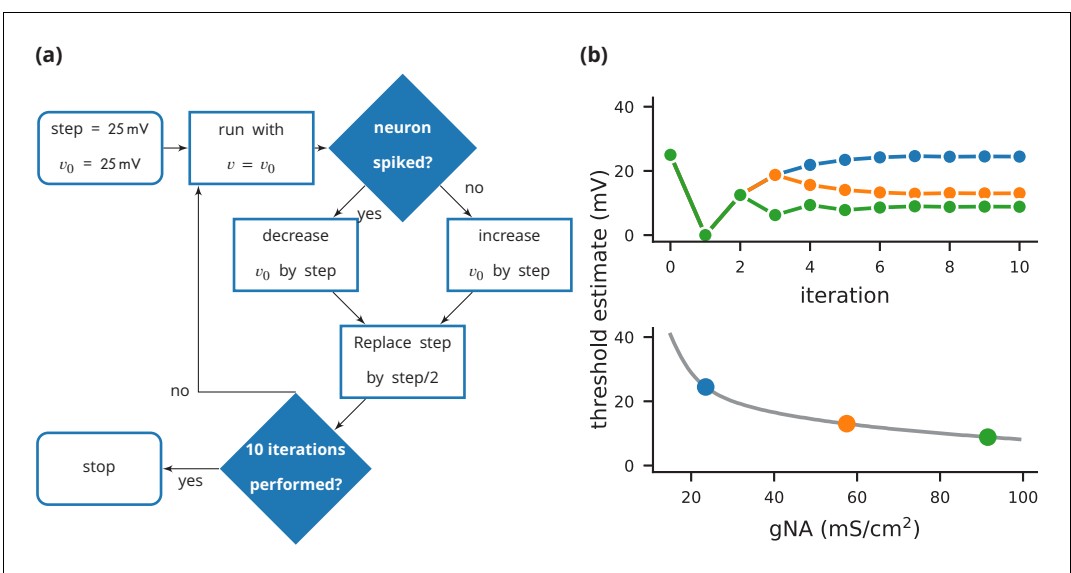

**Figure 4.** Case study 3: Using bisection to find a neuron's voltage threshold. (a) Schematic of the bisection algorithm for finding a neuron's voltage threshold. The algorithm is applied in parallel for different values of sodium density. (b) Top: Refinement of the voltage threshold estimate over iterations for three sodium densities (blue: 23.5 mS cm$^{-2}$, orange: 57.5 mS cm$^{-2}$, green: 91.5 mS cm$^{-2}$); Bottom: Voltage threshold estimation as a function of sodium density.

DOI: https://doi.org/10.7554/eLife.47314.007

```
1   from brian2 import *
2   defaultclock.dt = 0.01*ms
3
4   El = 10.613*mV; ENa = 115*mV; EK = -12*mV
5   gl = 0.3*mS/cm**2; gK = 36*mS/cm**2; C = 1*uF/cm**2
6   gNa_min = 15*mS/cm**2; gNa_max = 100*mS/cm**2
7
8   eqs = '''dv/dt = (gl*(El - v) + gNa*m**3*h*(ENa - v) + gK*n**4*(EK - v)) / C : volt
9            gNa : siemens/meter**2
10           dm/dt = alpham*(1 - m) - betam*m : 1
11           dn/dt = alphan*(1 - n) - betan*n : 1
12           dh/dt = alphah*(1 - h) - betah*h : 1
13           alpham = (0.1/mV)*(-v + 25*mV)/(exp((-v + 25*mV)/(10*mV)) - 1)/ms : Hz
14           betam = 4 * exp(-v/(18*mV))/ms : Hz
15           alphah = 0.07 * exp(-v/(20*mV))/ms : Hz
16           betah = 1/(exp((-v+30*mV) / (10*mV)) + 1)/ms : Hz
17           alphan = (0.01/mV) * (-v+10*mV) / (exp((-v+10*mV) / (10*mV)) - 1)/ms : Hz
18           betan = 0.125*exp(-v/(80*mV))/ms : Hz'''
19  neurons = NeuronGroup(100, eqs, threshold='v > 50*mV', method='exponential_euler')
20  neurons.gNa = 'gNa_min + (gNa_max - gNa_min)*1.0*i/N'
21  neurons.v = 0*mV
22  neurons.m = '1/(1 + betam/alpham)'
23  neurons.n = '1/(1 + betan/alphan)'
24  neurons.h = '1/(1 + betah/alphah)'
25  S = SpikeMonitor(neurons)
26
27  store()
28
29  # We locate the threshold by bisection
30  v0 = 25*mV*ones(len(neurons))
31  step = 25*mV
32
33  for i in range(10):
34      restore()
35      neurons.v = v0
36      run(20*ms)
37      v0[S.count == 0] += step
38      v0[S.count > 0] -= step
39      step /= 2.0
```

**Figure 5.** Case study 3: Simulation code to find a neuron's voltage threshold, implementing the bisection algorithm detailed in *Figure 4a*. The code simulates 100 unconnected neurons with sodium densities between 15 mS cm$^{-2}$ and 100 mS cm$^{-2}$, following the model of *Hodgkin and Huxley (1952)*. Results from these simulations are shown in *Figure 4b*.

DOI: https://doi.org/10.7554/eLife.47314.008

## Case study 3: Threshold finding

In this case study, we want to determine the voltage firing threshold of a neuron (*Figure 4*), modelled with three conductances, a passive leak conductance and voltage-dependent sodium and potassium conductances (*Figure 5*, l. 4–24).

To get an accurate estimate of the threshold, we use a bisection algorithm (*Figure 4a*): starting from an initial estimate and with an initial step width (*Figure 5*, l. 30–31), we set the neuron's membrane potential to the estimate (l. 35) and simulate its dynamics for 20 ms (l. 36). If the neuron spikes, that is if the estimate was above the neuron's threshold, we decrease our estimate (l. 38); if the neuron does not spike, we increase it (l. 37). We then halve the step width (l. 39) and perform the same process again until we have performed a certain number of iterations (l. 33) and converged to a precise estimate (*Figure 4b* top). Note that the order of operations is important here. When we

modify the variable $v$ in lines 37–38, we use the output of the simulation run on line 36, and this determines the parameters for the next iteration. A purely declarative definition could not represent this essential feature of the computational experiment.

For each iteration of this loop, we restore the network state (`restore()`; l. 34) to what it was at the beginning of the simulation (`store()`; l. 27). This `store()`/`restore()` mechanism is a key part of Brian's design for allowing computational experiments to be easily and flexibly expressed in Python, as it gives a very effective way of representing common computational experimental protocols. Examples that can easily be implemented with this mechanism include a training/testing/validation cycle in a synaptic plasticity setting; repeating simulations with some aspect of the model changed but the rest held constant (e.g. parameter sweeps, responses to different stimuli); or simply repeatedly running an identical stochastic simulation to evaluate its statistical properties.

At the end of the script, by performing this estimation loop in parallel for many neurons, each having a different maximal sodium conductance, we arrive at an estimate of the dependence of the voltage threshold on the sodium conductance (*Figure 4b* bottom).

## Comparison to other approaches

Such a simulation protocol could be implemented in other simulators as well, since they use a general programming language to control the simulation flow (e.g. SLI or Python for NEST; HOC or Python for NEURON) in similar ways to Brian. However, general simulation workflows are not part of description languages like NeuroML2/LEMS. While a LEMS model description can include a `<Simulation>` element, this is only meant to specify the duration and step size of one or several simulation runs, together with information about what variables should be recorded and/or displayed. General workflows, for example deciding whether to run another simulation based on the results of a previous simulation, are beyond its scope. These could be implemented in a separate script in a different programming language.

## Implementation level
### Case study 4: Real-time audio

The case studies so far were described by equations and algorithms on a level that is independent of the programming language and hardware that will eventually perform the computation. However, in some cases this lower level cannot be ignored. To demonstrate this, we will consider the example presented in *Figure 6*. We want to record an audio signal with a microphone and feed this signal— in real-time—into a neural network performing a crude 'pitch detection' based on the autocorrelation of the signal (*Licklider, 1962*). This model first transforms the continuous stimulus into a sequence of spikes by feeding the stimulus into an integrate-and-fire model with an adaptive threshold (*Figure 7*, l. 36–41). It then detects periodicity in this spike train by feeding it into an array of coincidence detector neurons (*Figure 6a*; *Figure 7*, l. 44–47). Each of these neurons receives the input spike train via two pathways with different delays (l. 49–51). This arrangement allows the network to detect periodicity in the input stimulus; a periodic stimulus will most strongly excite the neuron where the difference in delays matches the stimulus' period. Depending on the periodicity present in the stimulus, for example for tones of different pitch (*Figure 6b* middle), different subpopulations of neurons respond (*Figure 6b* bottom).

To perform such a study, our simulator has to meet two new requirements: firstly, the simulation has to run fast enough to be able to process the audio input in real-time. Secondly, we need a way to connect the running simulation to an audio signal via low-level code.

The challenge is to make the computational efficiency requirement compatible with the requirement of flexibility. With version 1 of Brian, we made the choice to sacrifice computational efficiency, because we reasoned that frequently in computational modelling, considerably more time was spent developing the model and writing the code than was spent on running it (often weeks versus minutes or hours; cf. *De Schutter, 1992*). However, there are obviously cases where simulation time is a bottleneck. To increase computational efficiency without sacrificing flexibility, We decided to make code generation the fundamental mode of operation for Brian 2 (*Stimberg et al., 2014*). Code generation was used previously in Brian 1 (*Goodman, 2010*), but only in parts of the simulation. This technique is now being increasingly widely used in other simulators, see *Blundell et al. (2018)* for a review.

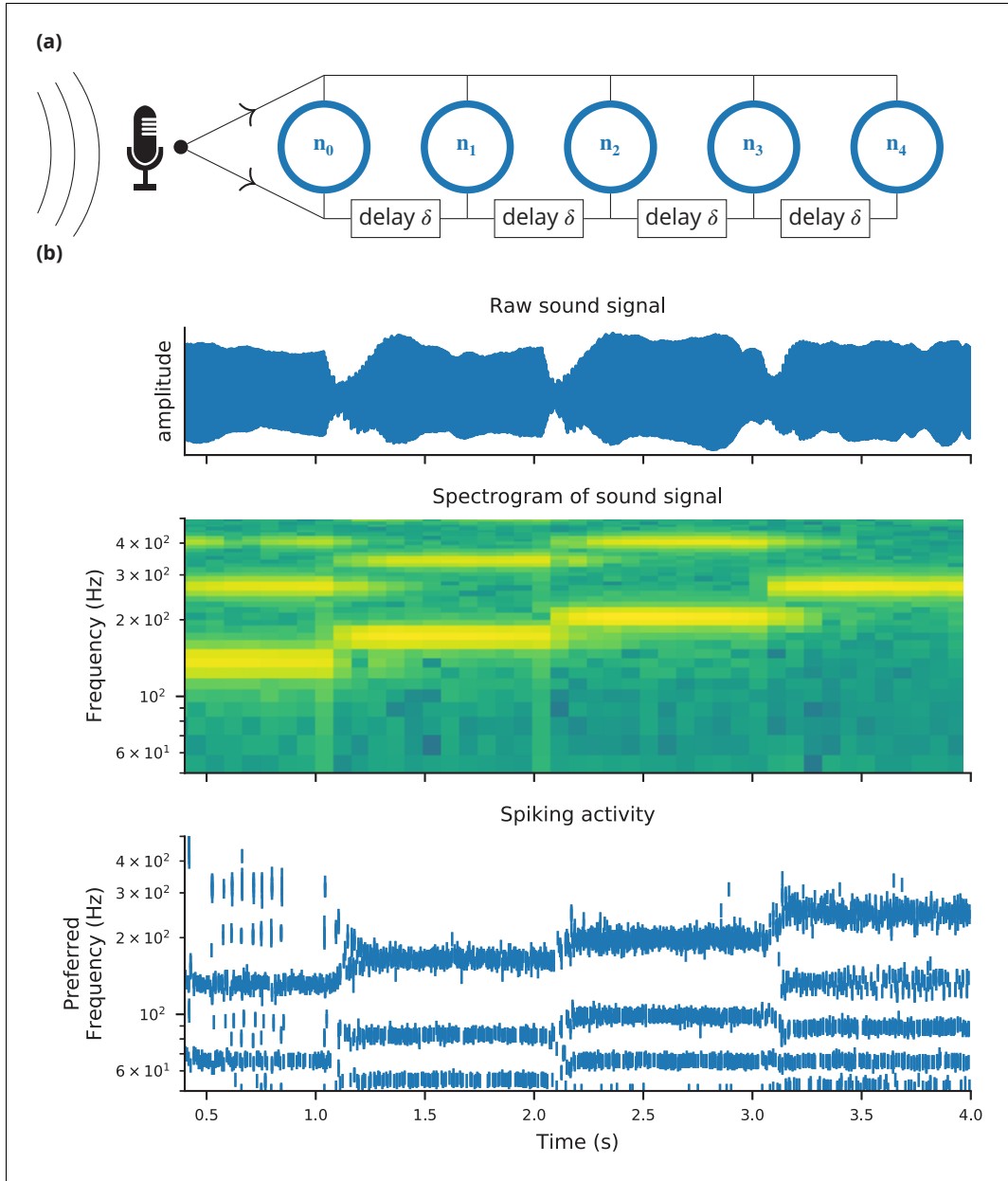

**Figure 6.** Case study 4: Neural pitch processing with real-time input. (**a**) Model schematic: Audio input is converted into spikes and fed into a population of coincidence-detection neurons via two pathways, one instantaneous, that is without any delay (top), and one with incremental delays (bottom). Each neuron therefore receives the spikes resulting from the audio signal twice, with different temporal shifts between the two. The inverse of this shift determines the preferred frequency of the neuron. (**b**) Simulation results for a sample run of the simulation code in *Figure 7*. Top: Raw sound input (a rising sequence of tones – C, E, G, C – played on a synthesised flute). Middle: Spectrogram of the sound input. Bottom: Raster plot of the spiking response of receiving neurons (group `neurons` in the code), ordered by their preferred frequency.
DOI: https://doi.org/10.7554/eLife.47314.009

In brief, from the high level abstract description of the model, we generate independent blocks of code (in C++ or other languages). We run these blocks in sequence to carry out the simulation. Typically, we first carry out numerical integration in one code block, check for threshold crossings in a second block, propagate synaptic activity in a third block, and finally run post-spike reset code in a fourth block. To generate this code, we make use of a combination of various techniques from

```
1   from brian2 import *
2   import os
3   set_device('cpp_standalone')
4
5   sample_rate = 48*kHz; buffer_size = 128; defaultclock.dt = 1/sample_rate
6   max_delay = 20*ms; tau_ear = 1*ms; tau_th = 5*ms
7   min_freq = 50*Hz; max_freq = 1000*Hz; num_neurons = 300; tau = 1*ms; sigma = .1
8
9   @implementation('cpp','''
10  PaStream *_init_stream() {
11      PaStream* stream;
12      Pa_Initialize();
13      Pa_OpenDefaultStream(&stream, 1, 0, paFloat32, SAMPLE_RATE, BUFFER_SIZE, NULL, NULL);
14      Pa_StartStream(stream);
15      return stream;
16  }
17
18  float get_sample(const double t) {
19      static PaStream* stream = _init_stream();
20      static float buffer[BUFFER_SIZE];
21      static int next_sample = BUFFER_SIZE;
22
23      if (next_sample >= BUFFER_SIZE)
24      {
25          Pa_ReadStream(stream, buffer, BUFFER_SIZE);
26          next_sample = 0;
27      }
28      return buffer[next_sample++];
29  }''', libraries=['portaudio'], headers=['<portaudio.h>'],
30      define_macros=[('BUFFER_SIZE', buffer_size),
31                     ('SAMPLE_RATE', sample_rate)])
32  @check_units(t=second, result=1)
33  def get_sample(t):
34      raise NotImplementedError('Use a C++-based code generation target.')
35
36  eqs_ear = '''dx/dt = (sound - x)/tau_ear: 1 (unless refractory)
37              dth/dt = (0.1*x - th)/tau_th : 1
38              sound = clip(get_sample(t), 0, inf) : 1 (constant over dt)'''
39  receptors = NeuronGroup(1, eqs_ear, threshold='x>th',
40                          reset='x=0; th = th*2.5 + 0.01',
41                          refractory=2*ms, method='exact')
42  receptors.th = 1
43
44  eqs_neurons = '''dv/dt = -v/tau+sigma*(2./tau)**.5*xi : 1
45                  freq : Hz (constant)'''
46  neurons = NeuronGroup(num_neurons, eqs_neurons, threshold='v>1', reset='v=0', method='euler')
47  neurons.freq = 'exp(log(min_freq/Hz)+(i*1.0/(num_neurons-1))*log(max_freq/min_freq))*Hz'
48
49  synapses = Synapses(receptors, neurons, on_pre='v += 0.5', multisynaptic_index='k')
50  synapses.connect(n=2)  # one synapse without delay; one with delay
51  synapses.delay['k == 1'] = '1/freq_post'
52
53  run(10*second)
```

**Figure 7.** Case study 4: Simulation code for the model shown in *Figure 6a*. The sound input is acquired in real time from a microphone, using user-provided low-level code written in C that makes use of an Open Source library for audio input (*Bencina and Burk, 1999*).
DOI: https://doi.org/10.7554/eLife.47314.010

symbolic mathematics and compilers that are available in third party Python libraries, as well as some domain-specific optimisations to further improve performance (see Appendix 1 for more details, or *Stimberg et al., 2014*; *Blundell et al., 2018*). We can then run the complete simulation in one of two modes, as follows.

In *runtime* mode, the overall simulation is controlled by Python code, which calls out to the compiled code objects to do the heavy lifting. This method of running the simulation is the default, because despite some computational overhead associated with repeatedly switching from Python to another language, it allows for a great deal of flexibility in how the simulation is run: whenever Brian's model description formalism is not expressive enough for a task at hand, the researcher can interleave the execution of generated code with a hand-written function that can potentially access and modify any aspect of the model. This facility is widely used in computational models using Brian.

In *standalone* mode, additional low-level code is generated that controls the overall simulation, meaning that during the main run of the simulation it is not necessary to switch back to Python. This gives an improvement to performance, but at the cost of reduced flexibility since we cannot translate arbitrary Python code into low level code. The standalone mode can also be used to generate code to run on a platform where Python is not available or not practical (such as a GPU; *Stimberg et al., 2018*).

The choice of which mode to use is left to the user, and will depend on details of the simulation and how much additional flexibility is required. The performance that can be gained from using the standalone mode also depends strongly on the details of the model; we will come back to this point in the discussion.

The second issue we needed to address for this case study was how to connect the running simulation to an audio signal via low-level code. The general issue here is how to extend the functionality of Brian. While Brian's syntax allows a researcher to define a wide range of models within its general framework, inevitably it will not be sufficient for all computational research projects. Taking this into account, Brian has been built with extensibility in mind. Importantly, it should be possible to extend Brian's functionality and still include the full description of the model in the main Python script, that is without requiring the user to edit the source code of the simulator itself or to add and compile separate modules.

As discussed previously, the runtime mode offers researchers the possibility to combine their simulation code with arbitrary Python code. However, in some cases, such as a model that requires real-time access to hardware (*Figure 6*), it may be necessary to add functionality at the target-language level itself. To this end, simulations can use a general extension mechanism: model code can refer not only to predefined mathematical functions, but also to functions defined in the target language by the user (*Figure 7*, l. 9–34). This can refer to code external to Brian, for example to third-party libraries (as is necessary in this case to get access to the microphone). In order to establish the link, Brian allows the user to specify additional libraries, header files or macro definitions (l. 29–31) that will be taken into account during the compilation of the code. With this mechanism the Brian simulator offers researchers the possibility to add functionality to their model at the lowest possible level, without abandoning the use of a convenient simulator and forcing them to write their model 'from scratch' in a low-level language. We think it is important to acknowledge that a simulator will never have every possible feature to cover all possible models, and we therefore provide researchers with the means to adapt the simulator's behaviour to their needs at every level of the simulation.

## Comparison to other approaches

The NEURON simulator can include user-written C code in `VERBATIM` blocks of an NMODL description, but there is no documented mechanism to link to external libraries. Another approach to interface a simulation with external input or output is to do this on the script level. For example, a recent study (*Dura-Bernal et al., 2017*) linked a NEURON simulation of the motor cortex to a virtual musculoskeletal arm, by running a single simulation step at a time, and then exchanging values between the two systems. The NEST simulator provides a general mechanism to couple a simulation to another system (e.g. another simulator) via the MUSIC interface (*Djurfeldt et al., 2010*). This framework has been successfully used to connect the NEST simulators to robotic simulators (*Weidel et al., 2016*). The MUSIC framework does support both spike-based and continuous interactions, but NEST cannot currently apply continuous-valued inputs as used here. Finally, model description languages such as NeuroML2/LEMS are not designed to capture this kind of interaction.

# Discussion

Brian 2 was designed to overcome some of the major challenges we saw for neural simulators (including Brian 1). Notably: the flexibility/performance dichotomy, and the need to integrate complex computational experiments that go beyond their neuronal and network components. As a result of this work, Brian can address a wide range of modelling problems faced by neuroscientists, as well as giving more robust and reproducible results and therefore contributing to increasing reproducibility in computational science. We now discuss these challenges in more detail.

Brian's code generation framework allows for a solution to the dichotomy between flexibility and performance. Brian 2 improves on Brian 1 both in terms of flexibility (particularly the new, very general synapse model; for more details see Appendix 5) and performance, where it performs similarly to simulators written in low-level languages which do not have the same flexibility (*Tikidji-Hamburyan et al., 2017*; also see section *Performance* below). Flexibility is essential to be useful for fundamental research in neuroscience, where basic concepts and models are still being actively investigated and have not settled to the point where they can be standardised. Performance is increasingly important, for example as researchers begin to model larger scale experimental data such as that provided by the Neuropixels probe (*Jun et al., 2017*), or when doing comprehensive parameter sweeps to establish robustness of models (*O'Leary et al., 2015*).

It is possible to write plugins for Brian to generate code for other platforms without modifying the core code, and there are several ongoing projects to do so. These include Brian2GeNN (*Stimberg et al., 2018*) which uses the GPU-enhanced Neural Network simulator (GeNN; *Yavuz et al., 2016*) to accelerate simulations in some cases by tens to hundreds of times, and Brian2CUDA (https://github.com/brian-team/brian2cuda). The modular structure of the code generation framework was designed for this in order to be ready for future trends in both high-performance computing and computational neuroscience research. Increasingly, high-performance scientific computing relies on the use of heterogeneous computing architectures such as GPUs, FPGAs, and even more specialised hardware (*Fidjeland et al., 2009*; *Richert et al., 2011*; *Brette and Goodman, 2012*; *Moore et al., 2012*; *Furber et al., 2014*; *Cheung et al., 2015*), as well as techniques such as approximate computing (*Mittal, 2016*). In addition to basic research, spiking neural networks may increasingly be used in applications thanks to their low power consumption (*Merolla et al., 2014*), and the standalone mode of Brian is designed to facilitate the process of converting research code into production code.

A neural computational model is more than just its components (neurons, synapses, etc.) and network structure. In designing Brian, we put a strong emphasis on the complete computational experiment, including specification of the stimulus, interaction with non-neuronal components, etc. This is important both to minimise the time and expertise required to develop computational models, but also to reduce the chance of errors (see below). Part of our approach here was to ensure that features in Brian are as general and flexible as possible. For example, the equations system intended for defining neuron models can easily be repurposed for defining non-neuronal elements of a computational experiment (case study 2, *Figure 3*). However, ultimately we recognise that any way of specifying all elements of a computational experiment would be at least as complex as a fully featured programming language. We therefore simply allow users to define these aspects in Python, the same language used for defining the neural components, as this is already highly capable and readable. We made great efforts to ensure that the detailed work in designing and implementing new features should not interfere with the goal that the user script should be a readable description of the complete computational experiment, as we consider this to be an essential element of what makes a computational model valuable.

Brian's approach to defining models leads to particularly concise code (*Tikidji-Hamburyan et al., 2017*), as well as code whose syntax matches closely descriptions of models in papers. This is important not only because it saves scientists time if they have to write less code, but also because such code is easier to verify and reproduce. It is difficult for anyone, the authors of a model included, to verify that thousands of lines of model simulation code match the description they have given of it. An additional advantage of the clean syntax is that Brian is an excellent tool for teaching, for example in the computational neuroscience textbook of *Gerstner et al. (2013)*. Expanding on this point, a major issue in computational science generally, and computational neuroscience in particular, is the reproducibility of computational models (*LeVeque et al., 2012*; *Eglen et al., 2017*;

*Podlaski et al., 2017*; *Manninen et al., 2018*). A frequent complaint of students and researchers at all levels, is that when they try to implement published models using their own code, they get different results. A fascinating and detailed description of one such attempt is given in *Pauli et al. (2018)*. These sorts of problems led to the creation of the ReScience journal, dedicated to publishing replications of previous models or describing when those replication attempts failed (*Rougier et al., 2017*). A number of issues contribute to this problem, and we designed Brian with these in mind. So, for example, users are required to write equations that are dimensionally consistent, a common source of problems. In addition, by requiring users to write equations explicitly rather than using pre-defined neuron types such as 'integrate-and-fire' and 'Hodgkin-Huxley', as in other simulators, we reduce the chance that the implementation expected by the user is different to the one provided by the simulator. We discuss this point further below, but we should note the opposing view that standardisation and common implementation are advantages has also been put forward (*Davison et al., 2008*; *Gleeson et al., 2010*; *Raikov et al., 2011*). Perhaps more importantly, by making user-written code simpler and more readable, we increase the chance that the implementation faithfully represents the description of a model. Allowing for more flexibility and targeting the complete computational experiment increases the chances that the entire simulation script can be compactly represented in a single file or programming language, further reducing the chances of such errors.

## Comparison to other approaches

We have described some of the key design choices we made for version 2 of the Brian simulator. These represent a particular balance between the conflicting demands of flexibility, ease-of-use, features and performance, and we now compare the results of these choices to other available options for simulations.

There are two main differences of approach between Brian and other simulators. Firstly, we require model definitions to be explicit. Users are required to give the full set of equations and parameters that define the model, rather than using 'standard' model names and default parameters (cf. *Brette, 2012*). This approach requires a slightly higher initial investment of effort from the user, but ensures that users know precisely what their model is doing and reduces the risk of a difference between the implementation of the model and the description of it in a paper (see discussion above). One limitation of this approach is that it makes it more difficult to design tools to programmatically inspect a model, for example to identify and shut down all inhibitory currents (although note that this issue remains for languages such as NeuroML and NineML that are primarily based on standard models as they include the ability to define arbitrary equations).

The second main difference is that we consider the complete computational experiment to be fundamental, and so everything is tightly integrated to the extent that an entire model can be specified in a single, readable file, including equations, protocols, data analysis, etc. In Neuron and NEST, model definitions are separate from the computational experiment script, and indeed written in an entirely different language (see below). This adds complexity and increases the chance of errors. In NeuroML and NineML, there is no way of specifying arbitrary computational experiments. One counter-argument to this approach is that clearly separating model definitions may reduce the effort in re-using models or programmatically comparing them (as in *Podlaski et al., 2017*).

A consequence of the requirement to make model definitions explicit, and an important feature for doing novel research, is that the simulator must support arbitrary user-specified equations. This is available in Neuron via the NMODL description format (*Hines and Carnevale, 2000*), and in a limited form in NEST using NESTML (*Plotnikov et al., 2016*). NeuroML and NineML now both include the option for specifying arbitrary equations, although the level of simulator support for these aspects of the standards is unclear. While some level of support for arbitrary model equations is now fairly widespread in simulators, Brian was the first to make this a fundamental, core concept that is applied universally. Some simulators that have since followed this approach include DynaSim (*Sherfey et al., 2018*), which is based on MATLAB, and ANNarchy (*Vitay et al., 2015*). Other new simulators have taken an alternative approach, such as Xolotl (*Gorur-Shandilya et al., 2018*) which is based on building hierarchical representations of neurons from a library of basic components. One aspect of the equation-based approach that is missing from other simulators is the specification of additional defining network features, such as synaptic connectivity patterns, in an equally flexible, equation-oriented way. Neuron is focused on single neuron modelling rather than networks, and

only supports directly setting the connectivity synapse-by-synapse. NEST, PyNN (*Davison et al., 2008*), NeuroML, and NineML support this too, and also include some predefined general connectivity patterns such as one-to-one and all-to-all. NEST further includes a system for specifying connectivity via a 'connection set algebra' (*Djurfeldt, 2012*) allowing for combinations of a few core types of connectivity. However, none have yet followed Brian in allowing the user to specify connectivity patterns via equations, as is commonly done in research papers.

## Performance

Running compiled code for arbitrary equations means that code generation must be used. This requirement leads to a problem: a simulator that makes use of a fixed set of models can provide hand-optimised implementations of them, whereas a fully flexible simulator must rely on automated techniques. By contrast, an advantage of automated techniques is that they can generate optimisations for specialisations of models. For example, using the CUBA benchmark (*Vogels and Abbott, 2005*; *Brette et al., 2007*) in which all neurons have identical time constants, Brian 2 is dramatically faster than Brian 1, NEURON and NEST (*Figure 8*, left). This happens because Brian 2 can generate a specialised optimisation of the code since the model definition states that time constants are the same. If instead we modify the benchmark to feature heterogeneous time constants (*Figure 8*, right), then Brian 2 has to do much more work since it can no longer use these optimisations, while the run times for NEST and NEURON do not change.

We can make two additional observations based on this benchmark. Firstly, the benefits of parallelisation via multi-threading depend heavily on the model being simulated. For a large homogeneous population, the single threaded and multi-threaded standalone runs of Brian 2 take approximately the same time, and the single threaded run is actually faster at smaller network sizes. For the heterogeneous population, the opposite result holds: multi-threaded is always faster at all network sizes.

The second observation is that the advantage of running a Brian 2 simulation in standalone mode is most significant for smaller networks, at least for the single threaded case (for the moment, multi-threaded code is only available for standalone mode).

It should be noted, however, that despite the fact that Brian 2 is the fastest simulator at large network sizes for this benchmark, this does not mean that Brian 2 is faster than other simulators such as NEURON or NEST in general. The NEURON simulator can be used to simulate the point neuron models used in this benchmark, but with its strong focus on the simulation of biologically detailed, multi-compartment neuron models, it is not well adapted to this task. NEST, on the other hand, has been optimised to simulate very large networks, with many synapses impinging on each neuron. Most importantly, Brian's performance here strongly benefits from its focus on running simulations on individual machines where all simulation elements are kept in a single, shared memory space. In contrast, NEST and NEURON use more sophisticated communication between model elements which may cost performance in benchmarks like the one shown here, but can scale up to bigger simulations spread out over multiple machines. For a fairly recent and more detailed comparison of simulators, see *Tikidji-Hamburyan et al. (2017)*, although note that they did not test the standalone mode of Brian 2.

## Limitations of brian

The main limitation of Brian compared to other simulators is the lack of support for running large networks over multiple machines, and scaling up to specialised, high-performance clusters as well as supercomputers. While this puts a limit on the maximum feasible size of simulations, the majority of neuroscientists do not have direct access to such equipment, and few computational neuroscience studies require such large scale simulations (tens of millions of neurons). More common is to run smaller networks but multiple times over a large range of different parameters. This 'embarrassingly parallel' case can be easily and straightforwardly carried out with Brian at any scale, from individual machines to cloud computing platforms or the non-specialised clusters routinely available as part of university computing services. An example for such a parameter exploration is shown in *Appendix 4—figure 2*. This simulation strongly benefits from parallelisation even on a single machine, with the simulation time reduced by about a factor of about 45 when run on a GPU.

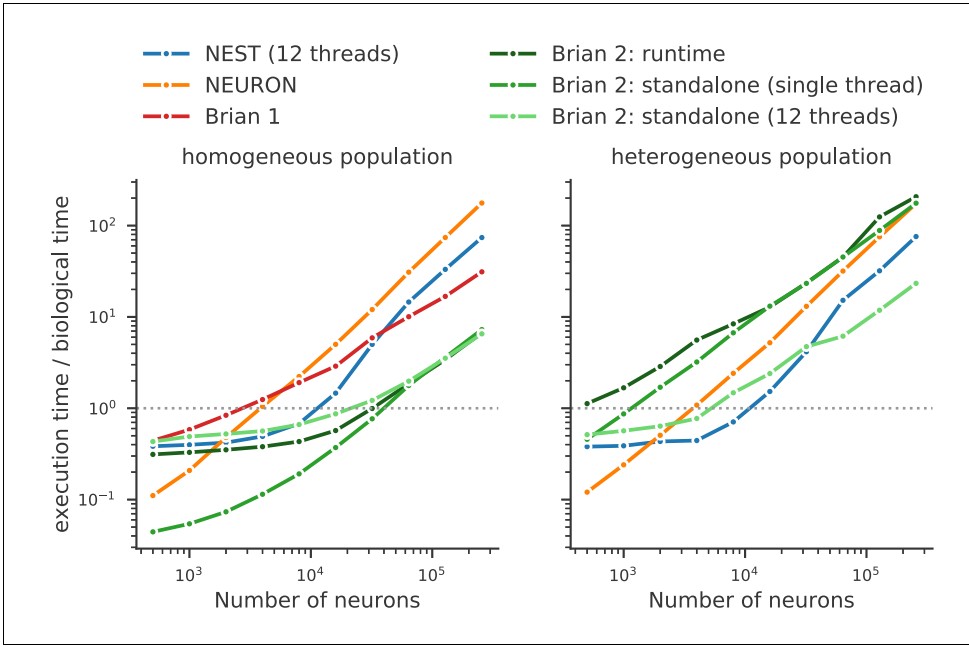

**Figure 8.** Benchmark of the simulation time for the CUBA network (***Vogels and Abbott, 2005***; ***Brette et al., 2007***), a sparsely connected network of leaky-integrate and fire network with synapses modelled as exponentially decaying currents. Synaptic connections are random, with each neuron receiving on average 80 synaptic inputs and weights set to ensure ongoing asynchronous activity in the network. The simulations use exact integration, but spike spike times are aligned to the simulation grid of 0.1 ms. Simulations are shown for a homogeneous population (left), where the membrane time constant, as well as the excitatory and inhibitory time constant, are the same for all neurons. In the heterogeneous population (right), these constants are different for each neuron, randomly set between 90% and 110% of the constant values used in the homogeneous population. Simulations were performed with NEST 2.16 (blue, ***Linssen et al., 2018***, RRID:SCR_002963), NEURON 7.6 (orange; RRID:SCR_005393), Brian 1.4.4 (red), and Brian 2.2.2.1 (shades of green, ***Stimberg et al., 2019b***, RRID:SCR_002998). Benchmarks were run under Python 2.7.16 on an Intel Core i9-7920X machine with 12 processor cores. For NEST and one of the Brian 2 simulations (light green), simulations made use of all processor cores by using 12 threads via the OpenMP framework. Brian 2 'runtime' simulations execute C++ code via the weave library, while 'standalone' code executes an independent binary file compiled from C++ code (see Appendix 1 for details). Simulation times do not include the one-off times to prepare the simulation and generate synaptic connections as these will become a vanishing fraction of the total time for runs with longer simulated times. Simulations were run for a biological time of 10 s for small networks (8000 neurons or fewer) and for 1 s for large networks. The times plotted here are the best out of three repetitions. Note that Brian 1.4.4 does not support exact integration for a heterogeneous population and has therefore not been included for that benchmark.
DOI: https://doi.org/10.7554/eLife.47314.011
The following source data is available for figure 8:

**Source data 1.** Benchmark results for the homogeneous network.
DOI: https://doi.org/10.7554/eLife.47314.012
**Source data 2.** Benchmark results for the heterogeneous network.
DOI: https://doi.org/10.7554/eLife.47314.013

Finally, let us note that this manuscript has focused exclusively on single-compartment point neuron models, where an entire neuron is represented without any spatial properties or compartmentalisation into dendrites, soma, and axon. Such models have been extensively used for the study of network properties, but are not sufficiently detailed for studying other questions, for example dendritic integration. For such studies, researchers typically investigate multi-compartment models, that is neurons modelled as a set of interconnected compartments. Currents across the membrane in each compartment are modelled in the same way as for point neurons, but there are additional axial currents with neighbouring compartments. Such models are the primary focus of simulators such as NEURON and GENESIS, but only have very limited support in simulators such as NEST.

While Brian is used mostly for point neurons, it does offer support for multi-compartmental models, using the same equation-based approach (see *Appendix 4—figure 1*). This feature is not yet as mature as those of specialised simulators such as NEURON and GENESIS, and is an important area for future development in Brian.

## Development and availability

Brian is released under the free and open CeCILL 2 license. Development takes place in a public code repository at https://github.com/brian-team/brian2 (*Brian contributors, 2019*). All examples in this article have been simulated with Brian 2 version 2.2.2.1 (*Stimberg et al., 2019b*). Brian has a permanent core team of three developers (the authors of this paper), and regularly receives substantial contributions from a number of students, postdocs and users (see Acknowledgements). Code is continuously and automatically checked against a comprehensive test suite run on all platforms, with almost complete coverage. Extensive documentation, including installation instructions, is hosted at http://brian2.readthedocs.org. Brian is available for Python 2 and 3, and for the operating systems Windows, OS X and Linux; our download statistics show that all these versions are in active use. More information can be found at http://briansimulator.org/.

## Acknowledgements

We thank the following contributors for having made contributions, big or small, to the Brian 2 code or documentation: Moritz Augustin, Victor Benichoux, Werner Beroux, Edward Betts, Daniel Bliss, Jacopo Bono, Paul Brodersen, Romain Cazé, Meng Dong, Guillaume Dumas, Ben Evans, Charlee Fletterman, Dominik Krzemiński, Kapil Kumar, Thomas McColgan, Matthieu Recugnat, Dylan Richard, Cyrille Rossant, Jan-Hendrik Schleimer, Alex Seeholzer, Martino Sorbaro, Daan Sprenkels, Teo Stocco, Mihir Vaidya, Adrien F Vincent, Konrad Wartke, Pierre Yger, Friedemann Zenke. Three of these contributors (CF, DK, KK) contributed while participating in Google's Summer of Code program.

## Additional information

### Funding

| Funder | Grant reference number | Author |
| --- | --- | --- |
| Agence Nationale de la Recherche | Axode ANR-14-CE13-0003 | Romain Brette |
| Royal Society | RG170298 | Dan FM Goodman |

The funders had no role in study design, data collection and interpretation, or the decision to submit the work for publication.

### Author contributions

Marcel Stimberg, Conceptualization, Software, Investigation, Visualization, Methodology, Writing—original draft, Writing—review and editing; Romain Brette, Conceptualization, Software, Supervision, Funding acquisition, Investigation, Methodology, Writing—original draft, Writing—review and editing; Dan FM Goodman, Conceptualization, Software, Supervision, Investigation, Methodology, Writing—original draft, Writing—review and editing

### Author ORCIDs

Marcel Stimberg (iD) https://orcid.org/0000-0002-2648-4790
Romain Brette (iD) https://orcid.org/0000-0003-0110-1623
Dan FM Goodman (iD) http://orcid.org/0000-0003-1007-6474

### Decision letter and Author response

Decision letter https://doi.org/10.7554/eLife.47314.027
Author response https://doi.org/10.7554/eLife.47314.028

## Additional files

### Supplementary files
• Source code 1. Benchmarking code.
DOI: https://doi.org/10.7554/eLife.47314.014
• Transparent reporting form
DOI: https://doi.org/10.7554/eLife.47314.015

### Data availability
Source code to replicate Figures 1-7, as well as the simulations shown in Appendix 4, are provided in a github repository (https://github.com/brian-team/brian2_paper_examples; copy archived at https://github.com/elifesciences-publications/brian2_paper_examples). Source code to run benchmarks as presented in Figure 8 is provided as a supplementary file together with this submission (Source code 1).

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

# Appendix 1

DOI: https://doi.org/10.7554/eLife.47314.016

## Design details

In this appendix, we provide further details about technical design decisions behind the Brian simulator. We also more exhaustively comment on the simulation code of the four case studies. Note that the example code provided as Jupyter notebooks (https://github.com/brian-team/brian2_paper_examples; *Stimberg et al., 2019a*; copy archived at https://github.com/elifesciences-publications/brian2_paper_examples) has extensive additional annotations as well.

## Mathematical level

### Physical units

Neural models are models of a physical system, and therefore variables have physical dimensions such as voltage or time. Accordingly, the Brian simulator requires quantities provided by the user, such as parameters or initial values of dynamical variables, to be specified in consistent physical units such as mV or s. This is in contrast to the approach of most other simulators, which simply define expected units for all model components, for example units of mV for the membrane potential. This is a common source of error because conventions are not always obvious and can be inconsistent. For example, while membrane surface area is often stated in units of $\mu m^2$, channel densities are often given in mS $cm^{-2}$. To remove this potential source of error, the Brian simulator enforces explicit use of units. It automatically takes care of conversions—multiplying a resistance (dimensions of $\Omega$) with a current (dimensions of A) will result in a voltage (dimensions of V)—and raises an error when physical dimensions are incompatible, for example when adding a current to a resistance. Unit consistency is also checked within textual model descriptions (e.g. *Figure 2*, l. 8–18) and variable assignments (e.g. l. 23–27). To make this possible, a dimension in SI units has to be assigned to each dimensional model variable in the model description (l. 8–18).

### Model dynamics

Neuron and synapse models are generally hybrid systems consisting of continuous dynamics described by differential equations and discrete events (*Brette et al., 2007*).

In the Brian simulator, differential equations are specified in strings using mathematical notation (*Figure 2*, l. 8–18). Differential equations can also be stochastic by using the symbol xi representing the noise term $\xi(t)$ (*Figure 3c*, l. 8). The numerical integration method can be specified explicitly, for example the pyloric circuit model chooses a second-order Runge-Kutta method (*Figure 2*, l. 22); without specification, an appropriate method is automatically chosen and reported. To this end, the user-provided equations are analysed symbolically using the Python package SymPy (*Meurer et al., 2017*), and transformed into a sequence of operations to advance the system's state by a single time step (for more details, see *Stimberg et al., 2014*).

This approach applies both to neuron models and to synaptic models. In many models, synaptic conductances do not need to be calculated for each synapse individually, instead they can be lumped into a single post-synaptic variable that is part of the neuronal model description. In contrast, non-linear synaptic dynamics as in the pyloric network example need to be calculated for each synapse individually. Using the same formalism as for neurons, the synaptic model equations can describe dynamics with differential equations (e.g. *Figure 2*, l. 31–32/44–47). Post-synaptic conductances or currents can then be calculated individually and summed up for each post-synaptic neuron as indicated by the (`summed`) annotation (l. 32 and 46).

Neuron- or synapse-specific values which are not updated by differential equations are also included in the string description. This can be used to define values that are updated by external mechanisms, for example the synaptic currents in each neuron (l. 15–16) are updated by the respective synapses (l. 32 and l. 46). The same mechanism can also be used for neuron-specific parameters such as the calcium target value (l. 17), or the label identifying the neuron type (l. 18). For optimisation, the flag (`constant`) can be added to indicate that the value will not change during a simulation.

Neural simulations typically refer to two types of discrete events: production of a spike, and reception of a spike. A spike is produced by a neuron when a certain condition on its variables is met. A typical case is the integrate-and-fire model, where a spike is produced when the potential reaches a threshold of a fixed value. But there are other cases when the condition is more complex, for example when the threshold is adaptive (*Platkiewicz and Brette, 2011*). To support conditions of all kind, Brian expects the user to define a mathematical expression as the `threshold`. In the case study 1, a spike is triggered whenever $v > -20mV$ (*Figure 2*, l. 21). No explicit resetting takes place, since the model dynamics describe the membrane potential trajectory during an action potential. For a simpler integrate-and-fire model as the one used in case study 2, the membrane potential is reset to a fixed value after the threshold crossing (*Figure 3*, l. 12). Such spike-triggered actions are most generally specified by providing one or more assignments and operations (reset) that should take place if the threshold condition is fulfilled; in the case study 1, this is mechanism is used to update the calcium trace (*Figure 2*, l. 21).

Once a spike is produced, it may affect variables of synapses and post-synaptic neurons (possibly after a delay). Again, this is specified generally as a series of assignments and operations. In the pyloric circuit example, this does not apply because the synaptic effect is continuous and not triggered by discrete spikes. In case study 2 (*Figure 3*) however, each spike has an instantaneous effect. For example, when a motoneuron spikes, the eye resting position is increased or decreased by a fixed amount. This is specified by `on_pre='x0_post += w'` (l. 15), where `on_pre` is a keyword for stating what operations should be executed when a pre-synaptic spike is received. These operations can refer to both local synaptic variables (here $w$, defined in the synaptic model) and variables of the pre- and postsynaptic neuron (here $x_0$, a variable of the post-synaptic neuron). In the same way, the `on_post` keyword can be used to specify operations executed when a postsynaptic spike is received, which allows defining various types of spike-timing-dependent models.

This general definition scheme applies to neurons and synapses, but as case study 2 (*Figure 3*) illustrates, it can also be used to define dynamical models of muscles and the environment. It also naturally extends to the modelling of non-neuronal elements of the brain such as glial cells (*Stimberg et al., 2019c*).

## Links between model components

The equations defining the dynamics of variables can only refer to other variables within the same model component, for example within the same group of neurons or synapses. Connections to other components have to be explicitly modelled using synaptic connections as explained above. However, we may sometimes also need to directly refer to the state of variables in other model component. For example, in case study 2 (*Figure 3*), the input to retinal neurons depends on eye and object positions, which are updated in a group separate from the group representing the retinal neurons (*Figure 3c*, l. 3–9). This can be expressed by defining a 'linked variable', which refers to a variable defined in a different model component. In the group modelling the retinal neurons, the variables `x_object` and `x_eye` are annotated with the (`linked`) flag to state that they are references to variables defined elsewhere (l. 23–24). This link is then made explicit by stating the group and variable they refer to via the `linked_var` function (l. 28–29).

## Initialisation

The description of its dynamics does not yet completely define a model, we also need to define its initial state. For some variables, this initial state can simply be a fixed value, for example in the pyloric network model, the neurons' membrane potential is initialised to the resting potential $v_r$ (**Figure 2** l. 24). In the general case, however, we might want to calculate the initial state; Brian therefore accepts arbitrary mathematical expressions for setting the initial value of state variables. These expressions can refer to model variables, as well as to pre-defined constant such as the index of a neuron within its group (i), or the total number of neurons within a group (N), as well as to pre-defined functions such as `rand()` (providing uniformly distributed random numbers between 0 and 1). In case study 1, we use this mechanism to initialise variables $w$ and $z$ randomly (**Figure 2**, l. 25–26); in case study 2, we assign individual preferred positions to each retinal neuron, covering the space from $-1$ to 1 in a regular fashion (**Figure 3c**, l. 30).

Mathematical expressions can also be used to select a subset of neurons and synapses and make conditional assignments. In case study 1, we assign a specific value to the conductance of synapses between ABPD and LP neurons by using the selection criterion `'label_pre == ABPD and label_post == LP'` (**Figure 2**, l. 36), referring to the custom label identifier of the pre- and post-synaptic neuron that has been introduced as part of the neuron model definition (l. 18). In this example there is only a single neuron per type, but the syntax generalises to groups of neurons of arbitrary size and is therefore preferable to the explicit use of numerical indices.

## Synaptic connections

The second main aspect of model construction is the creation of synaptic connections. For maximal expressivity, we again allow the use of mathematical expressions to define rules of connectivity. For example, in case study 1, following the schematic shown in **Figure 1a**, we would like to connect neurons with fast glutamatergic synapses according to two rules: 1) connections should occur between all groups, but not within groups of the same neuron type; 2) there should not be any connections from PY neurons to AB/PD neurons. We can express this with a string condition following the same syntax that we used to set initial values for synaptic conductances earlier (**Figure 2**, l. 35):

```
fast.connect('label_pre!=label_post and not (label_pre == PY
and label_post == ABPD)')
```

For more complex examples, in particular connection specifications based on the spatial location of neurons, see **Stimberg et al. (2014)**.

For larger networks, it can be wasteful to check a condition for each possible connection. Brian therefore also offers the possibility to use a mathematical expression to directly specify the projections of each neuron. In the eye movement example, each retinal neuron on the left hemifield (i.e. $x_{neuron}<0$) should connect to the first motoneuron (index 0), while neurons on the right hemifield (i.e. $x_{neuron}>0$) should connect to the second motoneuron (index 1). We can express this connection scheme by defining $j$, the postsynaptic target index, for each presynaptic neuron accordingly (with the `int` function converting a truth value into 0 or 1):

```
sensorimotor_synapses.connect(j='int(x_neuron_pre>0)')
```

This syntax can also be extended to generate more than one post-synaptic target per presynaptic neuron, using a syntax borrowed from Python's generator syntax (**Hettinger, 2002**, see the Brian 2 documentation at http://brian2.readthedocs.io for more details) These mechanisms can also be used to define stochastic connectivity schemes, either by specifying a fixed connection probability that will be evaluated in addition to the given conditions, or by specifying a connection probability as a function of pre- and post-synaptic properties.

Specifying synaptic connections in the way presented here has several advantages over alternative approaches. In contrast to explicitly enumerating the connections by referring to pre- and post-synaptic neuron indices, the use of mathematical expressions transparently conveys the logic behind the connection pattern and automatically scales with the size of the connected groups of neurons. These advantages are shared with simulators that provide pre-

defined connectivity patterns such as 'one-to-one' or 'all-to-all'. However, such approaches are not as general—for example they could not concisely define the connectivity pattern shown in *Figure 1a*—and can additionally suffer from ambiguity. For example, should a group of neurons that is 'all-to-all' connected to itself form autapses or not (cf. *Crook et al., 2012*)?

## Computational experiment level

The Brian simulator allows the user to write complete experiment descriptions that include both the model description and the simulation protocol in a single Python script as exemplified by the case studies in this article. In this section, we will discuss how the Brian simulator interacts with the statements and programming logic expressed in the surrounding script code.

### Simulation flow

In the case study 3, we use a specific simulation workflow, an iterative approach to finding a parameter value (*Figure 4a*). Many other simulation protocols are regularly used. For example, a simulation might consist of several consecutive runs, where some model aspect such as the external stimulation changes between runs. Alternatively, several different types of models might be tested in a single script where each is run independently. Or, a non-deterministic simulation might be run repeatedly to sample its behaviour. Capturing all these potential protocols in a single descriptive framework is hopeless, we therefore need the flexibility of a programming language with its control structures such as loops and conditionals.

Brian offers two main facilities to assist in implementing arbitrary simulation protocols. Simulations can be continued at their last state, potentially after activating/deactivating model elements, or changing global or group-specific constants and variables as shown above. Additionally, simulations can revert back to a previous state using the functions `store` and `restore` provided by Brian. In the example script shown in *Figure 5*, this mechanism is used to reset the network to an initial state after each iteration. The same mechanism allows for more complex protocols by referring to multiple states, for example to implement a train/test/validate protocol in a synaptic plasticity setting.

Providing explicit support for this functionality is not only a question of convenience; while the user could approximate this functionality by storing and resetting the systems state variables (membrane potentials, gating variables, etc.) manually, some model aspects such as action potentials that have not yet triggered synaptic effects (due to synaptic delays) are not easily accessible to the user.

### Model component scheduling

During each time step of a simulation run, several operations have to be performed. These include the numerical integration of the state variables, the propagation of synaptic activity, or the application of reset statements for neurons that emitted an action potential. All these operations have to be executed in a certain order. The Brian simulator approaches this issue in a flexible and transparent way: each operation has an associated clock with a certain time granularity dt, as well as a 'scheduling slot' and a priority value within that slot. Together, these elements determine the order of all operations across and within time steps.

By default, all objects are associated with the same clock, which simplifies setting a global simulation timestep for all objects (*Figure 5*, l. 2). However, individual objects may chose a different timestep, for example to record synaptic weights only sporadically during a long-running simulation run. In the same way, Brian offers a default ordering of all operations during a time step, but allows to change the schedule that is used, or to reschedule individual objects to other scheduling slots.

This amount of flexibility might appear to be unnecessary at a first glance and indeed details of the scheduling are rarely reported when describing models in a publication. Still, subtle differences in scheduling can have significant impact on simulation results (see *Appendix 2—figure 1* for an illustration). This is most obvious when investigating paradigms

such as spike-timing-dependent-plasticity with a high sensitivity to small temporal differences (*Rudolph and Destexhe, 2007*).

### Name resolution

Model descriptions refer to various 'names', such as variables, constants, or functions. Some of these references, such as function names or global constants, will have the same meaning everywhere. Others, such as state variables or neuron indices, will depend on the context. This context is defined by the model component, that is the group of neurons or the set of synapses, to which the description is attached. For example, consider the assignment to $g_{Na}$ (the maximum conductance of the sodium channel) in *Figure 5* (l. 20). Here, `gNa_min` and `gNa_max` refer to global constants (defined in l. 6; Brian also offers an alternative system where global constants and functions are explicitly provided via a Python dictionary instead of being deduced from values defined in the execution environment, but this system will not be further discussed here), whereas `i`, the neuron index, is a vector of values with one value for each neuron, and `N` refers to the total number of elements in the respective group.

It is important to note that the context is also given by its position in the program flow. For example, if we want to set the initial value for the gating variable $m$ to its steady value, then this value will depend on the membrane potential $v$ via the expressions for $\alpha_m$ and $\beta_m$. The order in which we set the values for $v$ and $m$ does therefore matter:

```
neuron.v=0*mV
neuron.m='1/(1+betam/alpham)'
```

While this might appear trivial, it shows how the procedural aspect of models, that is the order of operations, can be important. A purely descriptive approach, for example stating initial values for all variables as part of the model equations, would not always be sufficient (however, in this specific case, setting $v$ to 0 mV is unnecessary, since Brian automatically assigns the value to all uninitialised variables).

Some Python statements are translated into code that is run immediately, for example initialising a variable or creating synapses. Others are translated into code that is run at a later time. For example, the code to numerically integrate differential equations is not run at the point where those equations are defined, but rather at the point when the simulation is run via a call to the `run()` function. In this case, any named constants referred to in the equations will use their value at the time that the `run()` function is called, and not the value at the time the equations are defined. This allows for that value to change between multiple calls to `run()`, which may be useful to switch between global behaviours. For example, a typical use case is running with no external input current for a certain time to allow a neuron to settle into its stationary state, and then running with the current switched on by just changing the value of a constant from zero to some nonzero value between two consecutive `run()` calls.

## Implementation level

### Code generation

In order to combine the flexibility and ease-of-use of high-level descriptions with the execution speed of low-level programming languages such as C, we employ a code generation approach (*Goodman, 2010*). This code generation consists of three steps. The textual model description will first be transformed into a 'code snippet'. The generation of such a code snippet requires various transformations of the provided model description: some syntax elements have to be translated (e.g. the use of the `**` operator to denote the power operation to a call to the `pow` function for C/C++), variables that are specific to certain neurons or synapses have to be properly indexed (e.g. a reset statement `v = -70*mV` has to be translated into a statement along the lines of `v[neuron_index] = -70*mV`, and finally sequences of statements have to be expressed according to the target language syntax (e.g. by adding a semicolon to the end of each statement for C/C++). In a second step, these code snippets will then be embedded into a predefined target-code template, specific to the respective computation performed by the code. For example, the user-provided description of an integrate-and-fire neuron's reset

would be embedded into a loop that iterates over all the neurons that emitted an action potential during the current time step. Finally, the code has to be compiled and executed, giving it access to the memory location that the code has to read and modify. For further details on this approach, see *Goodman (2010)*; *Stimberg et al., 2014*.

## Code optimisation

Code resulting from the procedure described above will not necessarily perform computations in the most efficient way. Brian therefore uses additional techniques to further optimise the code for performance. Consider for example the $x$ variable—representing the receptor activity—in *Figure 7*, described by the differential equation in l. 36. This equation can be integrated analytically, and the above described code generation process would therefore generate code like the following (here presented as 'pseudo-code'):

```
for each neuron:
    x_new = sound + exp(-dt/tau_ear) * (sound - x_old)
```

However, the expression that is calculated for every neuron contains `exp(-dt /tau_ear)` which is not only identical for all neurons but also relatively costly to evaluate. Brian will identify such constant expressions, and calculate them only once outside of the loop:

```
c = exp(-dt/tau_ear)
for each neuron:
    x_new = sound + c * (sound - x_old)
```

In addition to this type of optimisation, the Brian simulator will also simplify arithmetic expressions, such as replacing `0*x` by `0`, or `x/x` by `1`. While all these optimizations could in principle also be performed by the programming-language compiler (e.g. gcc), we have found that performing these changes before handing over the code to the compiler led to bigger and more reliable performance benefits.

## Code execution: runtime mode

After the code generation process, each model component has been transformed into one or more 'code objects', each performing a specific computational task. For example, a group of integrate-and-fire neurons would typically result in three code objects. The first would be responsible for integrating the state variables over a single timestep, the second for checking the threshold condition to determine which neurons emit an action potential, and the third for applying the reset statements to those neurons. By default, these code objects will be executed in Brian's 'runtime mode', meaning that the simulation loop will be executed in Python and then call each of the code objects to perform the actual computation (in the order defined by the scheduling as described in the previous section). Note that while the code objects will typically be based on generated C++ code, they can be compiled and executed from within Python using binding libraries such as *weave* (formerly part of S*ciPy*; *Jones et al., 2001*) or *Cython* (*Behnel et al., 2011*).

This 'mixed' approach to model execution leaves the simulation control to the main Python process while the actual computations are performed in compiled code, operating on shared memory structures. This results in a considerable amount of flexibility: whenever Brian's model description formalism is not expressive enough for a task at hand, the researcher can interleave the execution of generated code with a hand-written function that can potentially access and modify any aspect of the model. In particular, such a function could intervene in the simulation process itself, for example by interrupting the simulation if certain criteria are met. The Jupyter notebook at https://github.com/brian-team/brian2_paper_examples contains an interactive version of case study 2 (*Figure 3*). In this example, the aforementioned mechanism is used to allow the user to interactively control a running Brian simulation, as well as for providing a graphical representation of the results that updates continuously.

While having all these advantages, the back-and-forth between the main loop in Python and the code objects also entails a performance overhead. This performance overhead takes a constant amount of time per code object and time step and does therefore matter less if the

individual components perform long-running computations, such as for large networks (see also *Figure 8*). On the other hand, for simulations of small or medium-sized networks, such as the network presented in case study 4 (*Figure 6*), this overhead can be considerable and the alternative execution mode presented in the following section might provide a better alternative.

## Code execution: standalone mode

As an alternative to the mode of execution presented in the previous section, the Brian simulator offers the so-called 'standalone mode', currently implemented for the C++ programming language. In this mode, Brian generates code that performs the simulation loop itself and executes the operations according to the schedule. Additionally, it creates code to manages the memory for all state variables and other data structures such as the queuing mechanism used for applying synaptic effects with delays. This code, along with the code of the individual code objects, establishes a complete 'standalone' version of the simulation run. When the resulting binary file is executed, it will perform the simulation and write all the results to disk. Since the generated code does not depend on any non-standard libraries, it can be easily transferred to other machines or architectures (e.g. for robotics applications). The generated code is free from any overhead related to Python or complex data structures and therefore executes with high performance.

For many models, the use of this mode only requires the researcher to add a single line to the simulation script (declaring `set_device('cpp_standalone')`), all aspects of the model descriptions, including assignments to state variables and the order of operations will be faithfully conserved in the generated code. The Python script will transparently compile and execute the standalone code, and then read the results back from disk so that the researcher does not have to adapt their analysis routines.

However, in contrast to the runtime execution mode presented earlier, it is not possible to interact with the simulation during its execution from within the Python script. In addition, certain programming logic is no longer possible, since all actions such as synapse generation or variable assignments are not executed when they are stated, but only as part of the simulation run.

In this execution mode, simulations of moderate size and complexity can be run in real-time (*Figure 8*), enabling studies such as the one presented in case study 4 (*Figure 6*). Importantly, this mode does not require the researcher to be actively involved in any details of the compilation, execution of the simulation or the retrieval of the results.

## Appendix 2

DOI: https://doi.org/10.7554/eLife.47314.016

### Simulation scheduling

(a)

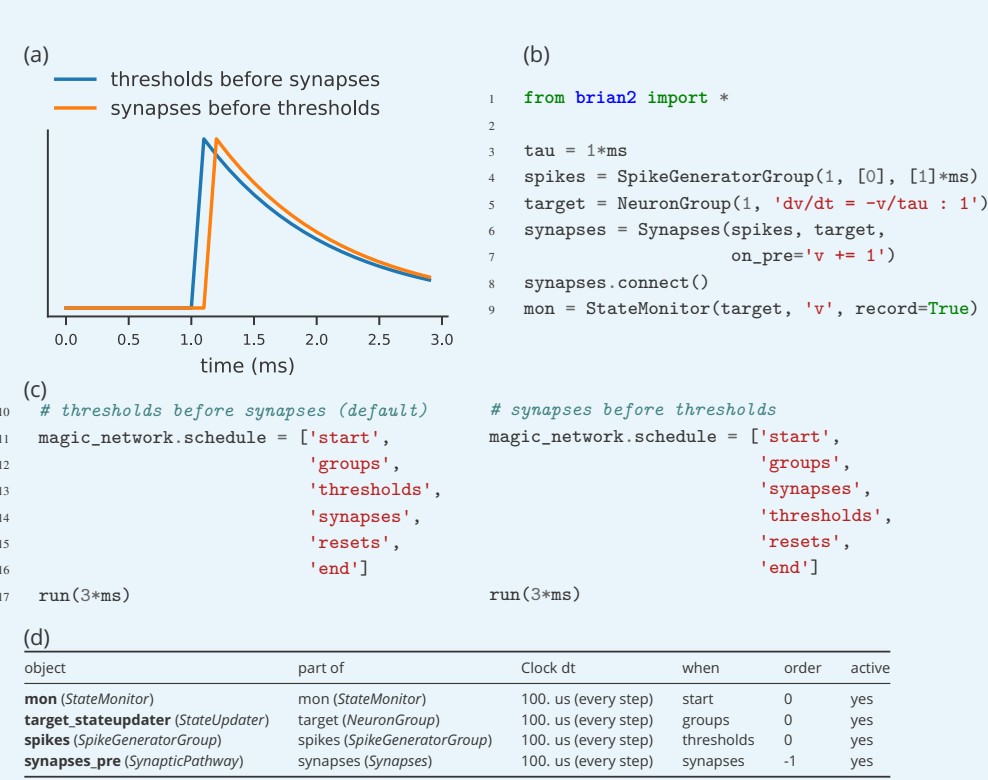

(b)

```
1   from brian2 import *
2
3   tau = 1*ms
4   spikes = SpikeGeneratorGroup(1, [0], [1]*ms)
5   target = NeuronGroup(1, 'dv/dt = -v/tau : 1')
6   synapses = Synapses(spikes, target,
7                       on_pre='v += 1')
8   synapses.connect()
9   mon = StateMonitor(target, 'v', record=True)
```

(c)

```
10  # thresholds before synapses (default)        # synapses before thresholds
11  magic_network.schedule = ['start',            magic_network.schedule = ['start',
12                            'groups',                                     'groups',
13                            'thresholds',                                 'synapses',
14                            'synapses',                                   'thresholds',
15                            'resets',                                     'resets',
16                            'end']                                        'end']
17  run(3*ms)                                      run(3*ms)
```

(d)

| object | part of | Clock dt | when | order | active |
|---|---|---|---|---|---|
| **mon** (*StateMonitor*) | mon (*StateMonitor*) | 100. us (every step) | start | 0 | yes |
| **target_stateupdater** (*StateUpdater*) | target (*NeuronGroup*) | 100. us (every step) | groups | 0 | yes |
| **spikes** (*SpikeGeneratorGroup*) | spikes (*SpikeGeneratorGroup*) | 100. us (every step) | thresholds | 0 | yes |
| **synapses_pre** (*SynapticPathway*) | synapses (*Synapses*) | 100. us (every step) | synapses | -1 | yes |

**Appendix 2—figure 1.** Demonstration of the effect of scheduling simulation elements. (**a**) Timing of synaptic effects on the post-synaptic cell for the two simulation schedules defined in (**c**). (**b**) Basic simulation code for the simulation results shown in (**a**). (**c**) Definition of a simulation schedule where threshold crossings trigger spikes and – assuming the absence of synaptic delays – their effect is applied directly within the same simulation time step (left; see blue line in (**a**)), and a schedule where synaptic effects are applied in the time step following a threshold crossing (right; see orange line in (**a**)). (**d**) Summary of the scheduling of the simulation elements following the default schedule (left code in (**c**)), as provided by Brian's `scheduling_summary` function. Note that for increased readibility, the objects from (**b**) have been explicitly named to match the variable names. Without this change, the code in (**b**) leads to the use of standard names for the objects (`spikegeneratorgroup, neurongroup, synapses,` and `statemonitor`).

DOI: https://doi.org/10.7554/eLife.47314.018

## Appendix 3

DOI: https://doi.org/10.7554/eLife.47314.016

### Model definitions in other simulation software

**(a)**

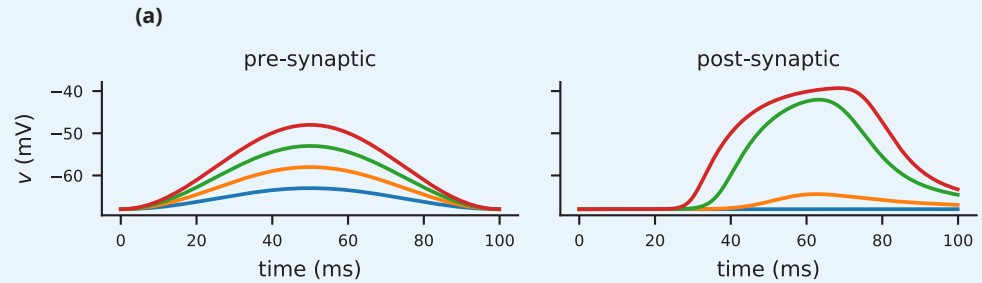

**(b) Brian code:**

```
eqs_slow = '''
I_slow_post = g_slow*m_slow*(v_post-E_syn) : amp (summed)
dm_slow/dt = k_1*(1-m_slow)/(1+exp(s_slow*(V_slow-v_pre)))
               - k_2*m_slow : 1 (clock-driven)
'''
slow_synapses = Synapses(circuit, circuit, model=eqs_slow, method='exact')
slow_synapses.connect('label_pre == ABPD and label_post != ABPD')
```

**(c) C++ code:**

```cpp
double get_ABLPsyn_G(double deltaT){
    m_ABLPsyn_inf=lookupsigmoid((V_ABLPsyn_thresh-V_mem[0])/V_ABLPsyn_slope);
    tau_ABLPsyn=(1.0-m_ABLPsyn_inf)*tau_ABLPsyn_diss;
    m_ABLPsyn=m_ABLPsyn+(m_ABLPsyn_inf-m_ABLPsyn)*deltaT/tau_ABLPsyn;
    return G_ABLPsyn_max*m_ABLPsyn;
}
//...
void update_model_neurons(double update_T)
{
//...
G_ABLPsyn_now=get_ABLPsyn_G(update_T);
V_ABLPsyn_inf=G_ABLPsyn_now*E_ABLPsyn;
//...
}
```

**Appendix 3—figure 1.** Graded synapse model. (**a**) Demonstration of the effect of the graded synapse model used in case study 1 (*Figure 1*, *Figure 2*). On the left, the membrane potential excursion of a pre-synaptic neuron is modelled by a squared sinusoidal function of time with varying amplitudes from 5 mV to 20 mV. The plot on the right shows the post-synaptic membrane potential of a cell receiving graded synaptic input from the pre-synaptic cell via the graded synapse model from case study 1 (slow cholinergic synapse, cf. *Golowasch et al., 1999*). The post-synaptic cell is modelled here as a simple leaky integrator with a single synaptic input current. (**b**) Code excerpt showing the Brian 2 definition of the graded synapse model used in (**a**), taken from the code used in case study 1 (*Figure 2*). (**c**) Code excerpt defining a graded synapse model in C++ as part of 'The Pyloric Network Model Simulator' (http://www.biology.emory.edu/research/Prinz/database-sensors/; *Günay and Prinz, 2010*). The complete code is 3510 lines.

DOI: https://doi.org/10.7554/eLife.47314.020

**(a) NeuroML2:**

```
<gradedSynapse id="gs" conductance="5pS" delta="5mV" Vth="-55mV"
 k="0.025per_ms" erev="0mV"/>
```

**LEMS:**

```
<ComponentType name="gradedSynapse" extends="baseGradedSynapse">
    <Property name="weight" dimension="none" defaultValue="1"/>
    
    
    
    
    
    <Exposure name="i" dimension="current"/>
    <Exposure name="inf" dimension="none"/>
    <Exposure name="tau" dimension="time"/>
    <Requirement name="v" dimension="voltage"/>
    <InstanceRequirement name="peer" type="baseGradedSynapse"/>
    <Dynamics>
        <StateVariable name="s" dimension="none"/>
        <DerivedVariable name="vpeer" dimension="voltage" select="peer/v"/>
        <DerivedVariable name="inf" dimension="none"
                         value="1/(1 + exp((Vth - vpeer)/delta))" exposure="inf"/>
        <DerivedVariable name="tau" dimension="time"
                         value="(1-inf)/k" exposure="tau"/>
        <DerivedVariable name="i" exposure="i"
                         value="weight * conductance * s * (erev-v)"/>
        <TimeDerivative variable="s" value="s_rate" />
    </Dynamics>
</ComponentType>
```

**(b) NEURON (NMODL):**

```
INDEPENDENT {t FROM 0 TO 1 WITH 1 (ms)}      BREAKPOINT {
NEURON {                                         SOLVE syn METHOD sparse
  POINT_PROCESS int1_lgsyn                       g = gmax *synon
  POINTER vpre                                   i = g*(v - e)
  RANGE gmax, g, e, i                          }
  NONSPECIFIC_CURRENT i                        KINETIC syn {
}                                                ~ synoff <-> synon (syninf(vpre)/tausyn(vpre),
UNITS {                                                    (1-syninf(vpre))/tausyn(vpre))
  (nA) = (nanoamp)                             }
  (mV) = (millivolt)                           INITIAL {
  (umho) = (micromho)                            synon = 0.0
}                                                synoff = 1.0
PARAMETER {                                     }
  gmax=0  (umho)                               FUNCTION syninf(v){
  e=0     (mV)                                    syninf = 1/(1+exp(-0.5*(v+49)))
  v       (mV)                                 }
}                                              FUNCTION tausyn(v){
STATE { synon synoff}                            tausyn = 2+98/(1+exp(-0.5*(v+49)))
ASSIGNED {                                     }
  i (nA)
  g (umho)
  vpre (mV)
}
```

**Appendix 3—figure 2.** Graded synapse model (cont.). (**a**) Definition of a graded synapse model in NeuroML2/LEMS. The graded synapse model as described in *Prinz et al. (2004)* has been added as a "core type" to the (not yet finalized) NeuroML2 standard and can therefore be accessed under the name gradedSynapse (top). It is fully defined via the LEMS definition partially reproduced here. (**b**) A definition of a graded synapse in the stomatogastric system, written in the NMODL language for the NEURON simulator. This model has been implemented in *Nadim et al. (1998)*, see https://senselab.med.yale.edu/ModelDB/showmodel.cshtml?model=3511.

DOI: https://doi.org/10.7554/eLife.47314.021

## Appendix 4

DOI: https://doi.org/10.7554/eLife.47314.016

# Additional Brian examples

## Multi-compartmental models

**(a)**

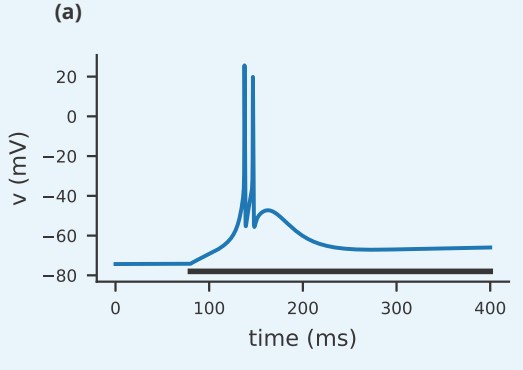

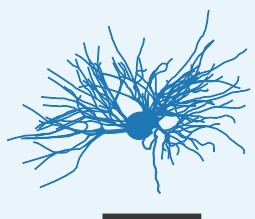

100 μm

**(b)**

```
1   # Constants
2   El = -76.5*mV; E_Na = 50*mV; E_K = -100*mV
3   # ...
4   eqs = Equations('''
5   Im = gl*(El-v) - I_Na - I_K - I_T: amp/meter**2
6   I_inj : amp (point current)
7
8   # HH-type currents for spike initiation
9   g_Na : siemens/meter**2
10  I_Na = g_Na * m**3 * h * (v-E_Na) : amp/meter**2
11  v2 = v - VT : volt  # shifted membrane potential (Traub convention)
12  dm/dt = (0.32*(mV**-1)*(13.*mV-v2)/
13          (exp((13.*mV-v2)/(4.*mV))-1.)*(1-m)-0.28*(mV**-1)*(v2-40.*mV)/
14          (exp((v2-40.*mV)/(5.*mV))-1.)*m) / ms * tadj_HH: 1
15  # ...
16  ''')
17  # Load morphology from SWC file
18  morpho = Morphology.from_file('tc200.CNG.swc')
19  neuron = SpatialNeuron(morpho, eqs, Cm=0.88*uF/cm**2, Ri=173*ohm*cm,
20                         method='exponential_euler')
21  # Only the soma has Na/K channels
22  neuron.main.g_Na = 100*msiemens/cm**2
23  neuron.main.g_K = 100*msiemens/cm**2
24  neuron.P_Ca = 1.7e-5*cm/second
25  # Distal dendrites
26  neuron.P_Ca['(distance + length/2) > 11*um'] = 8.5e-5*cm/second
27  neuron.v = -74*mV
28  neuron.m_T = 'm_T_inf'
29  neuron.h_T = 'h_T_inf'
30  mon = StateMonitor(neuron, ['v'], record=morpho[0])  # Record at soma
```

**Appendix 4—figure 1.** A multi-compartment model of a thalamic relay cell. (**a**) Simulation of a thamalic relay cell with increased T-current in distal dendrites (partially reproducing Figure 9C from **Destexhe et al. (1998)**. The plot shows the somatic membrane potential for a current injection of 75 pA during the period marked by the black line. The model consists of a total of 1291 compartments and is based on the morphology available on NeuroMorpho.Org

(*Ascoli et al., 2007*) under the ID NMO_00881, displayed on the right. This morphology is a reconstruction of a cell in the rat's ventrobasal complex, originally described in *Huguenard and Prince (1992)*. (**b**) Selected lines from the simulation code implementing the model shown in (**a**), focussing on the differences to single-compartmental models (as shown in case studies 1–4).

DOI: https://doi.org/10.7554/eLife.47314.023

## Parameter exploration

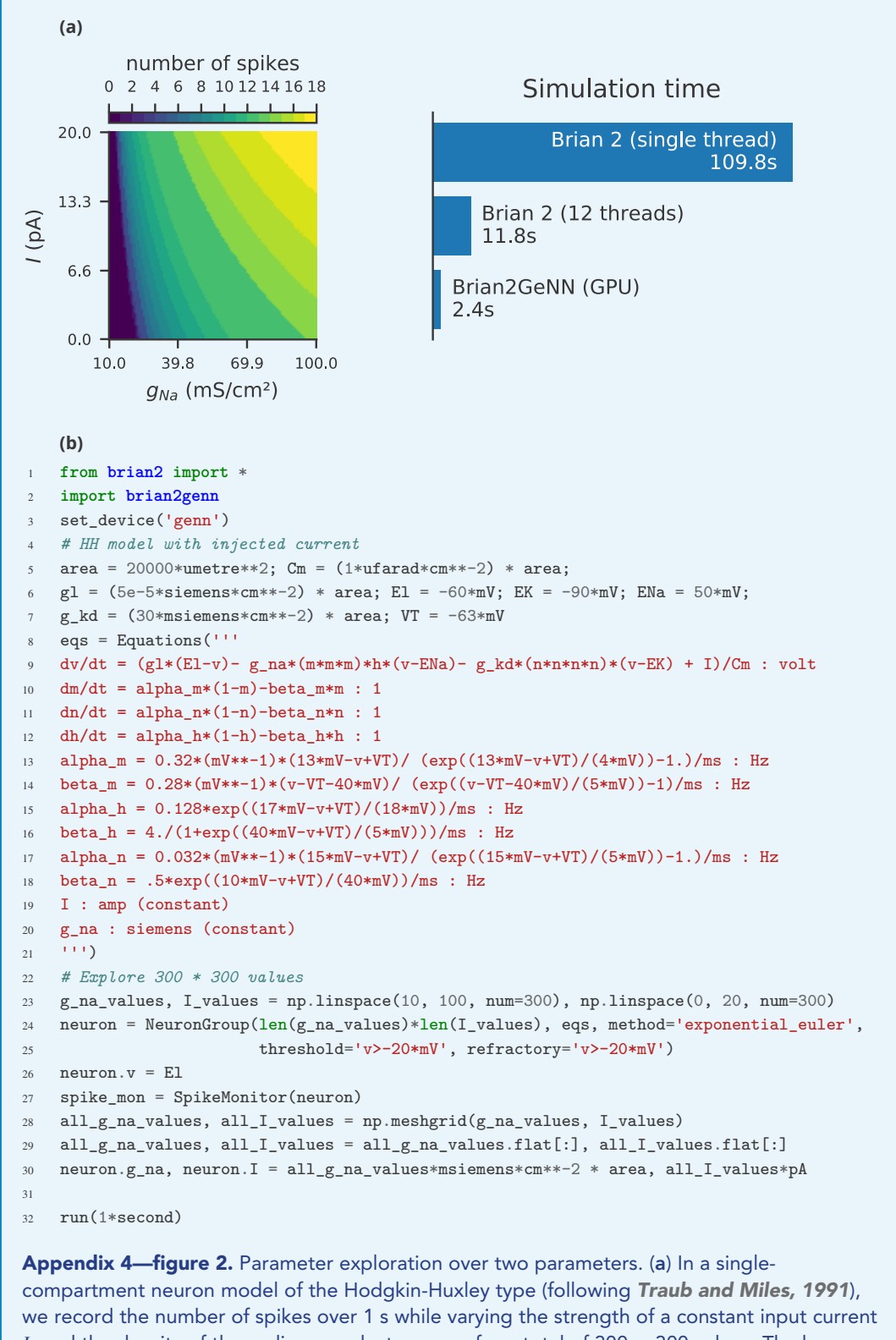

**(a)**

number of spikes

0 2 4 6 8 10 12 14 16 18

Simulation time

Brian 2 (single thread)
109.8s

Brian 2 (12 threads)
11.8s

Brian2GeNN (GPU)
2.4s

**(b)**

```
1   from brian2 import *
2   import brian2genn
3   set_device('genn')
4   # HH model with injected current
5   area = 20000*umetre**2; Cm = (1*ufarad*cm**-2) * area;
6   gl = (5e-5*siemens*cm**-2) * area; El = -60*mV; EK = -90*mV; ENa = 50*mV;
7   g_kd = (30*msiemens*cm**-2) * area; VT = -63*mV
8   eqs = Equations('''
9   dv/dt = (gl*(El-v)- g_na*(m*m*m)*h*(v-ENa)- g_kd*(n*n*n*n)*(v-EK) + I)/Cm : volt
10  dm/dt = alpha_m*(1-m)-beta_m*m : 1
11  dn/dt = alpha_n*(1-n)-beta_n*n : 1
12  dh/dt = alpha_h*(1-h)-beta_h*h : 1
13  alpha_m = 0.32*(mV**-1)*(13*mV-v+VT)/ (exp((13*mV-v+VT)/(4*mV))-1.)/ms : Hz
14  beta_m = 0.28*(mV**-1)*(v-VT-40*mV)/ (exp((v-VT-40*mV)/(5*mV))-1)/ms : Hz
15  alpha_h = 0.128*exp((17*mV-v+VT)/(18*mV))/ms : Hz
16  beta_h = 4./(1+exp((40*mV-v+VT)/(5*mV)))/ms : Hz
17  alpha_n = 0.032*(mV**-1)*(15*mV-v+VT)/ (exp((15*mV-v+VT)/(5*mV))-1.)/ms : Hz
18  beta_n = .5*exp((10*mV-v+VT)/(40*mV))/ms : Hz
19  I : amp (constant)
20  g_na : siemens (constant)
21  ''')
22  # Explore 300 * 300 values
23  g_na_values, I_values = np.linspace(10, 100, num=300), np.linspace(0, 20, num=300)
24  neuron = NeuronGroup(len(g_na_values)*len(I_values), eqs, method='exponential_euler',
25                       threshold='v>-20*mV', refractory='v>-20*mV')
26  neuron.v = El
27  spike_mon = SpikeMonitor(neuron)
28  all_g_na_values, all_I_values = np.meshgrid(g_na_values, I_values)
29  all_g_na_values, all_I_values = all_g_na_values.flat[:], all_I_values.flat[:]
30  neuron.g_na, neuron.I = all_g_na_values*msiemens*cm**-2 * area, all_I_values*pA
31
32  run(1*second)
```

**Appendix 4—figure 2.** Parameter exploration over two parameters. (**a**) In a single-compartment neuron model of the Hodgkin-Huxley type (following *Traub and Miles, 1991*), we record the number of spikes over 1 s while varying the strength of a constant input current $I$, and the density of the sodium conductance $g_{Na}$ for a total of $300 \times 300$ values. The bars on the right show the simulation time on the same machine used for *Figure 8* when using Brian 2's C++ standalone mode with a single thread (top), with 12 threads (middle), or when simulating it on a NVIDIA GeForce RTX 2080 Ti graphics card via the Brian2GeNN (*Stimberg et al., 2018*) interface to the GeNN (*Yavuz et al., 2016*) simulator (bottom). (**b**)

Code for the simulation shown in (**a**), here configured to run on the GPU via the Brian2GeNN interface (l.2-3).

DOI: https://doi.org/10.7554/eLife.47314.024

## Appendix 5

DOI: https://doi.org/10.7554/eLife.47314.016

### Comparison of Brian 1 and Brian 2

Brian 2 was rewritten from scratch, however it was designed to match the syntax of Brian 1 as closely as possible, breaking compatibility only when essential. Upgrading scripts from Brian 1 to Brian 2 is therefore usually straightforward. A detailed guide is available in the online documentation at https://brian2.readthedocs.io/en/stable/introduction/brian1_to_2/index.html.

### New features

#### Code generation

The major change from Brian 1 to Brian 2 is that all simulation objects are now based around code generation with behaviour determined by user-specified strings in standard mathematical notation. From these strings, C++ code is generated, compiled and run automatically. Brian 2 can be used in runtime mode (similar to Brian 1 but with individual objects accelerated using code generation), or standalone mode (in which a complete C++ source tree is generated which can be used independently of Brian and Python). Third party packages can extend this support to generate code for different devices, such as GPUs (e.g. *Stimberg et al., 2018*). New features have been added to make it easier to write code that can make use of and extend this code generation system, including extending functions by providing their definitions in a target language, and the `run_regularly()` method that covers much of what was previously done with the (still existing) `@network_operation` but allowing for code generation.

#### Equations

Brian always allowed users to write equations and differential equations in standard mathematical notation. Stochastic differential equations are now handled in a general way. In Brian 1, only additive noise was allowed and integrated with an Euler scheme. Brian 2 additionally supports multiplicative noise with the Heun and Milstein integration schemes. Numerical integration schemes can be added by the user using a general syntax. Variable time step integration using the GNU Scientific Library was added. Flags can now be added to equations to modify their behaviour (e.g. deactivating specific equations while the neuron is refractory, declaring values to be constant over a time step or run to enable optimisations).

#### Neurons

In addition to the new equations features above, the threshold, reset and refractoriness properties of neurons have now been greatly expanded. In Brian 1, these were handled by custom Python classes and could not easily be combined in complex ways. In Brian 2, each is defined by a string written in standard mathematical notation, determining a condition to evaluate (for threshold or refractoriness) or series of operations to be executed (for reset). Whether or not a neuron is refractory is stored in the (user accessible) `not_refractory` variable that is used alongside the `unless refractory` flag of the differential equations to switch off dynamics for user selected variables of refractory neurons. This structure allows for much greater flexibility and can be used with code generation.

#### Multi-compartmental modelling

Brian 1 had very basic support for modelling of neurons with a small number of compartments. Brian 2 adds support for detailed morphologies and specific integration schemes, see Appendix 4.

## Synapses

In the first release of Brian, synaptic connectivity was defined by the Connection class, which only allowed a single weight variable which was added to a target neuron variable when a pre-synaptic neuron fired. Later, a more general `Synapses` class was added which greatly expanded the flexibility, but was inefficient due to the lack of comprehensive support for code generation in Brian 1. This `Synapses` class is now the only mechanism in Brian 2, generalises the version from Brian 1 and adds code generation support. Synapses allows for a user-specified set of differential equations and parameters (exactly the same as for neurons) along with a specification of what operations should be calculated on the event of a pre- or post-synaptic spike. Multiple pathways with different delays are supported. Synapses can modify pre- or post-synaptic neurons in a discrete or continuous manner (to allow for more complex synapse models or rate-based models). Synapses can also target other synapses (for models of astrocytes for example, *Stimberg et al., 2019c*). Multiple synapses per neuron pair are now supported. Brian 1's Connection supported defining connectivity by an explicit array, or by specifying full, random, or one-to-one connectivity. Brian 2's `Synapses` generalises these with string-based arguments, and adds support for conditional connectivity (a string based expression determining which pairs to connect) and a generator-based syntax that allows you to write code similar to a for loop but that gets converted into efficient low-level code.

## Events

In Brian 1, there was only one type of event. A neuron created a spike event if a variable crossed a threshold, and this spike event triggered a reset on the source neuron, as well as synaptic activity. In Brian 2, these events remain but the user can also specify arbitrary events and triggered operations.

## Monitors

Brian 1 had a large collection of monitors to record different types of activity. These have been replaced by just three generalised versions that record discrete, continuous or population activity.

## Store/restore

Brian 2 has a new `store()` and `restore()` mechanism that saves and loads the entire simulation state.

## String-based indexing and evaluation

Brian 2 allows variables to be indexed by strings as well as numerical indices. For example, writing `G.z['x > 0'] = 'sin(y)'` would set the value of variable $z$ to $\sin(y)$ (where $y$ can be a single value or neuron variable), but only for those neurons where the variable $x>0$.

## Units

In Brian 1, only scalar values could have physical dimensions. Now arrays can also have units. In addition, consistency of dimensions is now used everywhere.

## Safety

A number of changes were made to minimise the chance that the user would write code that behaved differently from what was expected. This includes raising an error or warning whenever there is any ambiguity.

## Python 3

Brian 1 was written only for Python 2. Brian 2 is available for Python 2 and 3.

## Continuous integration

Brian 2 has a large test suite that is automatically run on multiple versions of Python, operating systems (Linux, Max, Windows), and architectures (32/64 bit). Installation has been improved to make it easier to install, and to ensure that C++ compiler tools are installed to make sure that the most high performance generated code can be used.

### Removed or replaced features

#### Packages

The aim of Brian 2 was to have a simpler, more flexible core package, and allow separate packages to provide extra functionality. The `brian.tools` package which provided some general purpose tools was therefore removed. The `brian.hears` package has been updated to `brian2hears` provided separately. An updated and generalised version of the `brian.modelfitting` (**Rossant et al., 2010**) package is in progress.

#### Library

Brian 1 featured a 'library' of models that could be used instead of writing equations explicitly. In line with the design philosophy of Brian 2 described in this paper, this feature was removed. All the equations are listed in the documentation and so Brian 1 models using these features can easily be updated.

#### STDP

Brian 1 had specific classes for STDP models. These are now obsolete as the `Synapses` class in Brian 2 covers everything they could do and more. Examples are given in the documentation of how to update code.

#### Connection class

The `Connection` class of Brian 1 has been removed in favour of the new `Synapses` class (see above).

