## [Decision Letter]

Thank you for submitting your article "Brian 2: an intuitive and efficient neural simulator" for consideration by *eLife*. Your article has been reviewed by three peer reviewers, and the evaluation has been overseen by a Reviewing Editor and Ronald Calabrese as the Senior Editor. The following individuals involved in review of your submission have agreed to reveal their identity: Scott Rich – Skinner Lab (Reviewer #1); Fleur Zeldenrust (Reviewer #2); Richard C Gerkin (Reviewer #3).

The reviewers have discussed the reviews with one another and the Reviewing Editor has drafted this decision to help you prepare a revised submission.

Summary:

The authors present Brian2, a flexible, extensible, simulator for neuroscience (or perhaps any dynamical system). Several features of Brian2 are presented by description, code samples, and case studies, and contrasted with alleged limitations of other approaches to simulation neuroscience. A foremost reason to use it is that it is intuitive and readable. Equations look like equations. This puts Brian in the tradition of tools like XPP that mathematical neuroscientists have relied upon but with the benefits of a modern approach to software development, user experience, etc.

While all reviewers appreciated this as a valuable, useful and powerful simulator tool for the computational neuroscience community, there were several concerns with the paper in its current form. Three essential revisions are detailed below.

It is noted that the reviewers encompassed those who are Brian users, casual Brian users, and non-Brian users. This was important in considering this work for *eLife* which is a more general and non-specialized journal.

Essential revisions:

1) The authors need to be explicit about the use and advantages of Brian2 over other simulators, so that its advantages (and differences) from other simulators are appreciated and realized. This needs to be presented without assuming that the user is already a Brian user and that it is already obvious that Brian2 may be superior to NEURON, NEST etc. Providing explicit examples and comparisons with other simulators would be useful. That is, explicitly state and show how and why Brian2 makes computational modeling available to a much wider group of researchers than before. For example, starting from some model equations, how would one proceed to use Brian2/python, DynaSim/MATLAB, XPP, NEST, NEURON. GENESIS etc. In this way, the "intuitive and efficient" aspects can be understood by the reader.

Further, by explicit comparisons and examples, the authors could bring forth why and how Brian2 is easy to use and helps make the researcher focus on model description rather than the intricate choices made in for instance writing ode solvers, which makes it also very useful for student projects. One can write one's equations and run them, so the math underlying the model is pretty clear and easy to work with. Show us why Brian2 is the better than other simulators.

Further specifics along these lines:

i) The authors have a tendency to "bury the lead" with regards to the key aspects of Brian 2 that make this simulation environment particularly useful. While this problem is present throughout the text, it is particularly evident in the Introduction. The Introduction reads more like a chronological explanation of the thought process behind Brian 2, rather than an argument for its importance and usefulness: indeed, key points are often lost in the middle of long paragraphs (for instance, the sentence in the Introduction section beginning "Thus, simulators generally have to find a trade-off…" is a key point that is not highlighted by the authors). I would suggest the authors consider reorganizing their Introduction, and to some degree the Discussion, in order to better highlight the key "thesis statements" of their manuscript.

ii) Throughout the "Design and Implementation" section, in particular with regards to the case studies, the authors make a variety of assertions regarding the ways in which their code improves upon Brian 1 and other simulation environments (NEURON, GENESIS, and NEST are highlighted in the Introduction). However, direct comparisons between Brian 2 and these existing environments are never made (and the areas in which Brian 2 improves over Brian 1 are highlighted sparsely). If, for example, the authors want to make the argument that Brian 2 is superior to its competition in defining "unconventional synapses", some discussion is necessary regarding why NEURON, GENESIS, NEST, etc. either cannot do what is presented in this paper or would do so in an inferior fashion. This issue remains the case for the multiple times in which the authors assert that Brian 2 is inherently more "flexible" or "readable" when compared to other simulators. For another key example, consider that the authors assert that "capturing all these potential protocols in a single declarative framework is impossible, but it can be easily expressed in a programming language with control structures such as loops and conditionals." This is a major assertion (as would any assertion that something is "impossible"), and at minimum requires some supporting evidence. Finally, while the presentation of direct code from Brian is very useful, these presentations are used in the manuscript to draw conclusions about this code's superiority to similar code in Brian 1 or other languages. As a reader, I cannot simply take the author's word that this is the case; instead, illustrative example code from the other simulators that perform same or similar tasks should be presented alongside the Brian code so that the reader can draw this conclusion for themselves.

iii) My main comment about the article is about the comparison with other simulators. This is only done in Figure 8, and this is a bit meager. As a computational (neuro)scientist, every time I want to make a model or simulation, I am faced with a choice of platforms. Which one I choose depends on how difficult the implementation is in the platform for this particular problem (i.e. NEST is good for large simple networks, but not for multi-compartment models, NEURON the other way around), but also on the performance on this platform (not only simulation time but also possibly memory problems). I do not expect a full benchmark of Brian 2 for simulation time and memory use for all possible platforms, but Figure 8 is a bit meager. Especially since for most computational neuroscientists the choice will not be between C++ and Brian, but between Brian and Nest, DynaSim/Matlab, Neuron, etc. So I think it would be nice if the article included some comparisons between these. Moreover, DynaSim (Sherfey et al., 2018) is not mentioned at all, even though the authors profile it as 'the Brian in Matlab', more or less. So I believe it should at least be discussed.

iv) There are several claims about the limitations of other approaches to modeling, especially a) the inability for these approaches to describe and run a simulation experiment protocol on a model and b) the inability to make equations explicit. This was especially noted as a weakness of declarative approaches to modeling (e.g. NeuroML/NineML). However, in NeuroML2, which has been in use for a number of years, both of these are possible. Perhaps some would find the experience of doing these through Brian 2 to be easier, but it is misleading to say that they cannot be done with declarative modeling approaches.

2) The authors need to be explicit about the intended audience and underlying assumptions regarding Brian 2. That is, in making the claim about Brian 2 being intuitive, generalizable etc., there would seem to be an assumption that the user is already a python expert? Similar to the point above, the authors need to provide guidance to a potential starting user.

Explicit comparison to other simulators, from ground zero, would be helpful.

For example, starting from some model equations, one could learn C/C++ code, or learn python to use Brian 2, or hoc to use NEURON, or MATLAB to use DynaSim etc. In doing this, the authors could bring about where Brian 2 could/should be used in comparison to other simulators. For example, NEURON is presumably superior to Brian 2 for multi-compartment models? Etc.

Again, the authors need to be explicit and clear about underlying and starting assumptions, so that the general reader/user can appreciate and understand Brian 2 to consider its usage.

Further specifics:

i) The intended audience for this article, and the corresponding style of writing, is often unclear or varies from section to section. Given the interdisciplinary audience of *eLife*, and the desire for this software to be used by a wide range of computational neuroscientists, I would imagine it is most desirable for this article to be accessible to readers that may not be true "experts" in computer science, or individuals for whom this wasn't their primary training. For this to be the case, more care needs to be taken in clearly defining some of the terminology that is used. One key example is the consistent use of the terminology "high-level" versus "low-level" with no clear definition as to what this means in the context of the paper. More importantly, one of the key features of Brian 2 appears to be the "technique of code generation", but there is no clear point at which this terminology is explicitly defined in a manner accessible to people not familiar with the concept.

ii) The interaction between Python and Brian 2 appears to be paramount to the argument made by the authors that Brian 2 represents a clear "step forward" for these type of simulation environments. However, in making this argument the authors make two major assumptions: A) anyone who will make use of Brian 2 will be inherently familiar with Python; B) the integration of Brian 2 and Python is what makes this simulation environment a major "step forward". However, assumption A is not necessarily the case (I, for example, have not used Python in my computational neuroscience research up to this point), and for those who fall under this category it is not at all clear that Brian 2 is more "readable" or "usable" than other environments (thus making assumption B questionable as well). These issues came forth in a couple of different fashions in my explorations of the Brian 2 code. A major pitfall was quite literally immediately apparent, as the online documentation regarding both Brian 2 and the "Jupyter Notebooks" used to demonstrate it was very much lacking; for example, nowhere in the documentation is there an explicit explanation of how to run a piece of Brian 2 code, or open a Jupyter Notebook. While this might be trivial to someone with an extensive background with Python, it is most certainly not for someone without this background. If the authors wish to assert that Brian 2 is inherently more "useable" than other simulation environments, these are things that can't be assumed, but must be explicitly stated, in the documentation; indeed, improved "usability" implies that this will be the case for all qualified computational neuroscientists, not just those with a particular background or skillset. As I went further with my exploration with Brian 2, I found many additional instances of terminology, code syntax, and other aspects of the programs that were not clear to me, and not explicitly explained or documented anywhere either in the Jupyter Notebooks or in the online documentation. All of this led me to the following conclusion: in order to make full use of Brian 2, I'd have to spend a significant period of time learning the underlying syntax and code structure, and it isn't clear to me that the time I would take to do this would be any less arduous or more streamlined than when I did similar things to use C code or NEURON.

3) Emphasis is placed on the performance of Brian 2 and its code-generation capabilities. While true high-performance computing simulations are beyond the scope of Brian 2, the embarrassingly parallel case should be trivial to handle (as noted by the authors), and GPU support is noted. However, I could find no examples of using GPUs in the online Brian 2 documentation. The authors should include examples in the documentation (and in supplementary materials) where hundreds of simulations of the same model (e.g. Izhikevich) are executed with different sets of parameters in parallel on the GPU (and dumps out the resulting waveforms).

---

## [Author Response]

Essential revisions:1) The authors need to be explicit about the use and advantages of Brian2 over other simulators, so that its advantages (and differences) from other simulators are appreciated and realized. This needs to be presented without assuming that the user is already a Brian user and that it is already obvious that Brian2 may be superior to NEURON, NEST etc. Providing explicit examples and comparisons with other simulators would be useful. That is, explicitly state and show how and why Brian2 makes computational modeling available to a much wider group of researchers than before. For example, starting from some model equations, how would one proceed to use Brian2/python, DynaSim/MATLAB, XPP, NEST, NEURON. GENESIS etc. In this way, the "intuitive and efficient" aspects can be understood by the reader.Further, by explicit comparisons and examples, the authors could bring forth why and how Brian2 is easy to use and helps make the researcher focus on model description rather than the intricate choices made in for instance writing ode solvers, which makes it also very useful for student projects. One can write one's equations and run them, so the math underlying the model is pretty clear and easy to work with. Show us why Brian2 is the better than other simulators.

We agree with the reviewers that some comparison of the modelling process with Brian compared to other simulators would be helpful for readers, and we have therefore added comparisons to other simulators. For the first case study (pyloric network), we have provided excerpts of equivalent code for the graded synapse model (in Appendix 3), in C++, NeuroML2/LEMS, and NEURON/NMODL, which we have taken from the literature. It cannot be implemented with NESTML – it could certainly be integrated in NEST, but not without detailed knowledge of the simulator codebase (i.e., it could be done by a NEST developer). In order to keep the paper to a reasonable length we restricted our comparisons to these simulators and languages, as a comprehensive approach for all simulators and case studies would make a detailed review paper in itself, and would be difficult without involving the authors of simulators. In addition to this detailed code comparison, we added a discussion of other simulators at the end of the text for each case study (focusing on the same set of simulators).

Further specifics along these lines:i) The authors have a tendency to "bury the lead" with regards to the key aspects of Brian 2 that make this simulation environment particularly useful. While this problem is present throughout the text, it is particularly evident in the Introduction. The Introduction reads more like a chronological explanation of the thought process behind Brian 2, rather than an argument for its importance and usefulness: indeed, key points are often lost in the middle of long paragraphs (for instance, the sentence in the Introduction section beginning "Thus, simulators generally have to find a trade-off…" is a key point that is not highlighted by the authors). I would suggest the authors consider reorganizing their Introduction, and to some degree the Discussion, in order to better highlight the key "thesis statements" of their manuscript.

We have rewritten parts of the Abstract, Introduction, design and implementation, and Discussion to address this point.

ii) Throughout the "Design and Implementation" section, in particular with regards to the case studies, the authors make a variety of assertions regarding the ways in which their code improves upon Brian 1 and other simulation environments (NEURON, GENESIS, and NEST are highlighted in the Introduction). However, direct comparisons between Brian 2 and these existing environments are never made (and the areas in which Brian 2 improves over Brian 1 are highlighted sparsely).

We have added these direct comparisons as discussed above. We also added a new appendix listing the key improvements of Brian 2 over Brian 1 (Appendix 5).

If, for example, the authors want to make the argument that Brian 2 is superior to its competition in defining "unconventional synapses", some discussion is necessary regarding why NEURON, GENESIS, NEST, etc. either cannot do what is presented in this paper or would do so in an inferior fashion.

As discussed above, we have added explicit comparisons of code for different simulators in Appendix 3. We also discuss it at the end of the presentation of case study 1.

This issue remains the case for the multiple times in which the authors assert that Brian 2 is inherently more "flexible" or "readable" when compared to other simulators. For another key example, consider that the authors assert that "capturing all these potential protocols in a single declarative framework is impossible, but it can be easily expressed in a programming language with control structures such as loops and conditionals." This is a major assertion (as would any assertion that something is "impossible"), and at minimum requires some supporting evidence.

This is rather a theoretical argument. If the language is powerful enough to express anything computable, then by definition it is a general-purpose programming language, and not a domain-specific declarative language. Therefore, to the extent that a declarative language (such as NeuroML) is not fully general (i.e. not Turing complete), there are protocols that it cannot express (a protocol being any kind of procedure for manipulating the model than can be algorithmically described). We have reworded this claim to make this argument clearer.

Finally, while the presentation of direct code from Brian is very useful, these presentations are used in the manuscript to draw conclusions about this code's superiority to similar code in Brian 1 or other languages. As a reader, I cannot simply take the author's word that this is the case; instead, illustrative example code from the other simulators that perform same or similar tasks should be presented alongside the Brian code so that the reader can draw this conclusion for themselves.

As discussed previously, we have added some explicit examples for the first case study in Appendix 3. Note that one of the motivations to show the Brian 2 code for each case study was to demonstrate that it is possible to fit the complete code in a single page, which is impossible for any other simulator that we have looked at.

iii) My main comment about the article is about the comparison with other simulators. This is only done in Figure 8, and this is a bit meager. As a computational (neuro)scientist, every time I want to make a model or simulation, I am faced with a choice of platforms. Which one I choose depends on how difficult the implementation is in the platform for this particular problem (i.e. NEST is good for large simple networks, but not for multi-compartment models, NEURON the other way around), but also on the performance on this platform (not only simulation time but also possibly memory problems). I do not expect a full benchmark of Brian 2 for simulation time and memory use for all possible platforms, but Figure 8 is a bit meager. Especially since for most computational neuroscientists the choice will not be between C++ and Brian, but between Brian and Nest, DynaSim/Matlab, Neuron, etc. So I think it would be nice if the article included some comparisons between these.

We have added comparisons to other simulators in terms of syntax (see points above), and added performance comparisons with Brian 1, NEST, and NEURON to Figure 8. This is a complex issue because it is very hard to do a fair performance comparison of different simulators without 1) testing a wide range of different models and 2) involving the original authors from each simulator being compared, to make sure that the comparison is fair. To do this properly would have to be the subject of its own detailed review paper. From our perspective, we have seen that a number of published papers that compare simulators made choices in their Brian implementations that dramatically affect performance. For example, in Tikidji-Hamburyan et al., 2017, which is mostly very well done, they did not use the standalone mode of Brian 2, which for their examples would have made the code run several times faster. We assume that similar issues are present for other simulators and that these results should therefore be interpreted with caution. While we are able to write our own examples to make sure they run efficiently in Brian, we could not guarantee that code we would write for other simulators was as efficient as it could possibly be. For this reason, we decided in our initial submission to include no performance comparisons with other simulators.

However, we do agree with the reviewers that readers are likely to be interested in such comparisons, and we have therefore done our best to add some to Figure 8 (for Neuron and NEST), with detailed explanations in the discussion, while also noting the limitations of these benchmarks. For Neuron and NEST we based our code on the reference scripts provided by the Neuron and NEST teams for Brette et al., 2007, as well as on examples from their documentation. We note that for these examples, Brian running in standalone mode compares very favourably to both Neuron and NEST, but as this is only one example, we do not wish to make a strong claim that Brian is more efficient.

We have also added some text about the limitations of Brian 2 relative to other simulators in the discussion. Specifically, Brian 2 is not designed to run over multiple machines and therefore is not adapted to the simulation of very large models, unlike NEST. There is some support for multicompartmental modelling in Brian 2 (see below), but clearly this is a more specific strength of NEURON and GENESIS.

Moreover, DynaSim (Sherfey et al., 2018) is not mentioned at all, even though the authors profile it as 'the Brian in Matlab', more or less. So I believe it should at least be discussed.

We have added this to our Discussion.

iv) There are several claims about the limitations of other approaches to modeling, especially a) the inability for these approaches to describe and run a simulation experiment protocol on a model and b) the inability to make equations explicit. This was especially noted as a weakness of declarative approaches to modeling (e.g. NeuroML/NineML). However, in NeuroML2, which has been in use for a number of years, both of these are possible. Perhaps some would find the experience of doing these through Brian 2 to be easier, but it is misleading to say that they cannot be done with declarative modeling approaches.

We do not claim that NeuroML and NineML are unable to make equations explicit, only that it cannot represent arbitrary simulation experiment protocols (as discussed above). As far as we can tell, neither NeuroML nor NineML include a mechanism for changing the parameters for subsequent simulation runs based on the outcome of the initial runs, for example.

2) The authors need to be explicit about the intended audience and underlying assumptions regarding Brian 2. That is, in making the claim about Brian 2 being intuitive, generalizable etc., there would seem to be an assumption that the user is already a python expert? Similar to the point above, the authors need to provide guidance to a potential starting user.Explicit comparison to other simulators, from ground zero, would be helpful.For example, starting from some model equations, one could learn C/C++ code, or learn python to use Brian 2, or hoc to use NEURON, or MATLAB to use DynaSim etc. In doing this, the authors could bring about where Brian 2 could/should be used in comparison to other simulators. For example, NEURON is presumably superior to Brian 2 for multi-compartment models? Etc.Again, the authors need to be explicit and clear about underlying and starting assumptions, so that the general reader/user can appreciate and understand Brian 2 to consider its usage.

We have added comparisons to other approaches and multi-compartmental modelling throughout the article and described these changes in our other responses to the reviewers.

On the specific issue of programming language choice, it is correct that some basic knowledge of Python is required to use Brian 2 (not expert; it is not necessary, for example, to know about object-oriented programming). All simulators require the user to know a language, at least for scripting. In fact, all the major simulators have now chosen to provide Python bindings, because it has become a standard language for scientific computing (MATLAB is indeed another major scientific language).

Therefore, the use of Python is not what distinguishes Brian 2 or makes it particularly intuitive. Our point is rather that the user is not required to know much more than this general language. Indeed, all other major simulators use an additional language to describe models; for example, to use NEURON one needs to master the scripting language (either hoc or Python) and in addition the domain language NMODL to describe ionic channel properties. In Brian 2, all models are defined within the script by their mathematical equations. Another very important aspect is that the user is not required to know how the models must be simulated (e.g. numerical integration), unlike when writing code in C++.

We have tried to clarify this point at the beginning of Design and Implementation.

Further specifics:i) The intended audience for this article, and the corresponding style of writing, is often unclear or varies from section to section. Given the interdisciplinary audience of eLife, and the desire for this software to be used by a wide range of computational neuroscientists, I would imagine it is most desirable for this article to be accessible to readers that may not be true "experts" in computer science, or individuals for whom this wasn't their primary training. For this to be the case, more care needs to be taken in clearly defining some of the terminology that is used. One key example is the consistent use of the terminology "high-level" versus "low-level" with no clear definition as to what this means in the context of the paper. More importantly, one of the key features of Brian 2 appears to be the "technique of code generation", but there is no clear point at which this terminology is explicitly defined in a manner accessible to people not familiar with the concept.

We have added a new paragraph defining high- and low-level at the beginning of Design and Implementation, and expanded the definition of code generation in the Introduction.

ii) The interaction between Python and Brian 2 appears to be paramount to the argument made by the authors that Brian 2 represents a clear "step forward" for these type of simulation environments. However, in making this argument the authors make two major assumptions: A) anyone who will make use of Brian 2 will be inherently familiar with Python; B) the integration of Brian 2 and Python is what makes this simulation environment a major "step forward". However, assumption A is not necessarily the case (I, for example, have not used Python in my computational neuroscience research up to this point), and for those who fall under this category it is not at all clear that Brian 2 is more "readable" or "usable" than other environments (thus making assumption B questionable as well). These issues came forth in a couple of different fashions in my explorations of the Brian 2 code. A major pitfall was quite literally immediately apparent, as the online documentation regarding both Brian 2 and the "Jupyter Notebooks" used to demonstrate it was very much lacking; for example, nowhere in the documentation is there an explicit explanation of how to run a piece of Brian 2 code, or open a Jupyter Notebook. While this might be trivial to someone with an extensive background with Python, it is most certainly not for someone without this background. If the authors wish to assert that Brian 2 is inherently more "useable" than other simulation environments, these are things that can't be assumed, but must be explicitly stated, in the documentation; indeed, improved "usability" implies that this will be the case for all qualified computational neuroscientists, not just those with a particular background or skillset. As I went further with my exploration with Brian 2, I found many additional instances of terminology, code syntax, and other aspects of the programs that were not clear to me, and not explicitly explained or documented anywhere either in the Jupyter Notebooks or in the online documentation. All of this led me to the following conclusion: in order to make full use of Brian 2, I'd have to spend a significant period of time learning the underlying syntax and code structure, and it isn't clear to me that the time I would take to do this would be any less arduous or more streamlined than when I did similar things to use C code or NEURON.

As explained above, the fact that Brian 2 uses Python rather than another language is not the main reason why Brian 2 is, in our view, more intuitive than other simulators. One of the main reasons is that the user is essentially required to know *only* Python. More specifically: 1) there is no difference between scripting language and model description language, unlike NEURON and NEST, and 2) the user is not required to specify the detailed implementation of the models, unlike with C/C++.

Regarding the first point, both NEURON and NEST now recommend using their Python scripting interfaces for new users rather than their own domain specific languages (HOC and SLI), because (a) it makes no sense for a new user to use a domain specific language if the same functionality is available in a general purpose language with good library support, development environments and debugging tools, and also (b) following a general trend in computational science towards the use of Python. However, models cannot be defined at the level of the Python script with either NEURON (which requires NMODL) or NEST (which requires NESTML, or C++). For the case of NEURON code, it seems unarguable that from the point of view of either a naïve or highly experienced user in all the languages used, a single Python file with model definitions (in standard mathematical notation), simulation control and analysis in one file and one language will be more readable than one written in two or three different languages (NMODL for model definition, HOC or Python for simulation control and perhaps Matlab for analysis).

Regarding the second point, to write your own custom C code for simulations in not only tedious, but it is also a dangerous practice in terms of reliability and reproducibility (see e.g. (Pauli et al., 2018) for a recent case study).

Finally on this point, we want to thank the reviewer for highlighting the lack of a good “quick start” guide for non-Python users in the Brian documentation, which we have now corrected (https://brian2.readthedocs.io/en/latest/introduction/scripts.html). However, to say that the statement “the online documentation … was very much lacking” based solely on the fact that we didn’t have a quick-start guide for non-Python users seems unreasonable. Firstly, in all of the thousands of messages to our email support group and conversations with users in person, this is the first time anyone has mentioned this as a problem, and good quick-start guides to Python and Jupyter exist online. Secondly, the documentation has been an enormous effort and is often mentioned to us by users as one of the main reasons that they learned Brian or switched to it from other simulators. It is extremely comprehensive, with tutorials, examples, a user guide, advanced guide and detailed reference documentation on every single feature, all cross-referenced, amounting to over 700 pages in printed form. In addition, in the online documentation, every single tutorial and example has a link allowing you to launch it, edit and run it in your browser without having to install anything locally, using the Binder service.

We have added a new paragraph at the beginning of Design and Implementation discussing the choice of programming language.

3) Emphasis is placed on the performance of Brian 2 and its code-generation capabilities. While true high-performance computing simulations are beyond the scope of Brian 2, the embarrassingly parallel case should be trivial to handle (as noted by the authors), and GPU support is noted. However, I could find no examples of using GPUs in the online Brian 2 documentation. The authors should include examples in the documentation (and in supplementary materials) where hundreds of simulations of the same model (e.g. Izhikevich) are executed with different sets of parameters in parallel on the GPU (and dumps out the resulting waveforms).

We have added an additional simulation to the Appendix (Appendix 4—figure 2), where we show code for the implementation of a parameter exploration. The example runs on GPU using Brian2GeNN, requiring only a single additional line of code (set_device(‘genn’)). GPU support for Brian is provided by external libraries and is not in the core of Brian 2, and we feel that a detailed discussion would be out of scope for this manuscript. We have therefore only added a brief comment on this in the Discussion.